# Helminth-induced Th2 cell dysfunction is distinct from exhaustion and is maintained in the absence of antigen

**Johanna A. Knipper** [1,2], **Alasdair Ivens** [1,2], **Matthew D. Taylor** [1,2]*

**1** Institute of Immunology and Infection Research, School of Biological Sciences, University of Edinburgh, Edinburgh, United Kingdom, **2** Centre for Immunity, Infection and Evolution, School of Biological Sciences, University of Edinburgh, Edinburgh, United Kingdom

* Matthew.Taylor@ed.ac.uk

**Data Availability Statement:** All gene array data is available in Gene Expression Omnibus (https://www.ncbi.nlm.nih.gov/geo/), accession number GSE114308. All other data are contained within the paper and its Supporting Information.

## Abstract

T cell-intrinsic regulation, such as anergy, adaptive tolerance and exhaustion, is central to immune regulation. In contrast to Type 1 and Type 17 settings, knowledge of the intrinsic fate and function of Th2 cells in chronic Type 2 immune responses is lacking. We previously showed that Th2 cells develop a PD-1/PD-L2-dependent intrinsically hypo-responsive phenotype during infection with the filarial nematode *Litomosoides sigmodontis*, denoted by impaired functionality and parasite killing. This study aimed to elucidate the transcriptional changes underlying Th2 cell-intrinsic hypo-responsiveness, and whether it represents a unique and stable state of Th2 cell differentiation. We demonstrated that intrinsically hypo-responsive Th2 cells isolated from *L. sigmodontis* infected mice stably retained their dysfunctional Th2 phenotype upon transfer to naïve recipients, and had a divergent transcriptional profile to classical Th2 cells isolated prior to hypo-responsiveness and from mice exposed to acute Type 2 stimuli. Hypo-responsive Th2 cells displayed a distinct transcriptional profile to exhausted CD4$^+$ T cells, but upregulated Blimp-1 and the anergy/regulatory-associated transcription factors Egr2 and c-Maf, and shared characteristics with tolerised T cells. Hypo-responsive Th2 cells increased mRNA expression of the soluble regulatory factors *Fgl2*, *Cd38*, *Spp1*, *Areg*, *Metrnl*, *Lgals3*, and *Csf1*, and a subset developed a T-bet$^+$IFN-γ$^+$ Th2/Th1 hybrid phenotype, indicating that they were not functionally inert. Contrasting with their lost ability to produce Th2 cytokines, hypo-responsive Th2 cells gained IL-21 production and IL-21R blockade enhanced resistance to *L. sigmodontis*. IL-21R blockade also increased the proportion of CD19$^+$PNA$^+$ germinal centre B cells and serum levels of parasite specific IgG1. This indicates a novel regulatory role for IL-21 during filarial infection, both in controlling protection and B cell responses. Thus, Th2 cell-intrinsic hypo-responsiveness is a distinct and stable state of Th2 cell differentiation associated with a switch from a classically active IL-4$^+$IL-5$^+$ Th2 phenotype, to a non-classical dysfunctional and potentially regulatory IL-21$^+$Egr2$^+$c-Maf$^+$Blimp-1$^+$IL-4$^{lo}$IL-5$^{lo}$T-bet$^+$IFN-γ$^+$ Th2 phenotype. This divergence towards alternate Th2 phenotypes during chronicity has broad implications for the outcomes and treatment of chronic Type 2-related infections and diseases.

**Funding:** This work was supported by the MRC UK (https://mrc.ukri.org) grant number MR/K020196/1, and the Wellcome Trust (https://wellcome.ac.uk/home) grant number 095831. The funders had no role in study design, data collection and analysis, decision to publish, or preparation of the manuscript.

**Competing interests:** The authors have declared that no competing interests exist.

## Author summary

Helminth parasites chronically infect over 1 billion people, with filarial nematodes accounting for 120 million. Protection is mediated by Th2 cells, but infection invokes dominant down-regulatory immune responses forming a major barrier to the development of protective immunity. We previously demonstrated that during murine filarial infection Th2 cells change phenotype, developing a novel form of Th2 cell-intrinsic dysfunction that impairs parasite clearance. Here we investigate the gene expression profile of Th2 cell dysfunction, and whether it represents a unique and stable form of T cell differentiation. Dysfunctional Th2 cells had a different gene expression profile to effector Th2 cells, and retained their unresponsive phenotype in the absence of antigen and active infection when transferred to a naïve recipient. Dysfunctional Th2 cells produced IL-21, through which they may inhibit protective immunity, as IL-21R neutralisation increased parasite killing. In comparison to known forms of T cell intrinsic regulation, dysfunctional Th2 cells had a distinct gene expression profile to exhausted T cells, but shared characteristics with anergy and tolerance. Together this indicates that Th2 cell dysfunction represents a unique and stable state of Th2 cell differentiation, which has important implications for understanding the outcomes of helminth infections and designing therapies for Th2-mediated allergies.

## Introduction

Human helminth infections are characterized by impaired Type 2 immune responses, the combined result of immune suppression by the parasite and immune regulation by the host [1]. This results in chronic infections in over 1 billion individuals worldwide, with protective immunity taking decades to develop [2]. Immune regulation during helminth infections, and immune tolerance to allergens, both associate with a change from a classical inflammatory Type 2 phenotype to a modified or tolerant Type 2 phenotype; denoted by increased IL-10 and a switch from inflammatory IgE to non-inflammatory IgG4 [1,3,4]. Thus, allergic inflammation and helminth infection share similar activation pathways, and in both cases immune regulation during chronicity associates with a change in Type 2 phenotype rather than a global downregulation of Type 2 responses. The mechanisms underlying this phenotype change are still largely unknown.

T cell-intrinsic regulation, e.g. exhaustion, anergy, and adaptive tolerance, is central to the development of immune tolerance [5], and the control of immune responses to infections and tumours [6]. The various forms of T cell-intrinsic unresponsiveness represent distinct states of T cell differentiation with unique functional properties and transcriptional signatures [5–8], and can lead to the development of tolerised or exhausted memory responses [8–10]. Thus, the development of T cell intrinsic regulation or dysfunction can have profound consequences on infection susceptibility and development of protective memory. Whilst a range of Th2 cell-extrinsic regulatory mechanisms have been characterised, e.g. regulatory T and B cells, and alternatively-activated macrophages [1,3,11,12], little is known of how Type 2 immune responses are regulated by intrinsic changes in Th2 cell function and fate.

There is some evidence for Th2 cell-intrinsic regulation during helminth infections. T cells from patients with chronic nematode infections show impaired TCR signalling [13,14], and suppression in human filariasis is associated with the upregulation of T cell anergy factors [15]. Th2 cells become intrinsically hypo-responsive or dysfunctional during murine

schistosomiasis [16], and exposure to *Fasciola hepatica* tegumental antigens induces an aner-gic-like T cell phenotype [17,18]. We demonstrated that Th2 cell-intrinsic hypo-responsive-ness develops during murine infection with the filarial nematode *Litomosoides sigmodontis*, and that blockade of the PD-1/PD-L2 pathway increases resistance and recovers Th2 function-ality [19,20]. Hypo-responsive Th2 cells in both schistosomiasis and filariasis show an intrinsi-cally-impaired ability to produce Th2 cytokines (IL-4, IL-5, IL-10, and IL-13) and to proliferate, both in the presence of antigen and mitogen [16,19,20]. Thus, immune regulation during chronic helminth infections associates with an intrinsically hypo-responsive or dys-functional Th2 cell phenotype, which influences both resistance and pathology.

It has not been established whether Th2 cell-intrinsic hypo-responsiveness represents a known form of T cell dysfunction or a hitherto undescribed differentiation state, and the underlying mechanisms are not well defined. The involvement of PD-1 suggests similarities with T cell exhaustion. However, in contrast with exhaustion, PD-1 acts in conjunction with PD-L2 rather than PD-L1 [19]. Susceptibility to *L. sigmodontis* associates with the recruitment of PD-L2$^+$ monocytes to the infection site that control Th2 cell functional quality, suggesting that monocytes may play a role in induction of Th2 cell-intrinsic hypo-responsiveness [21]. In murine schistosomiasis, Th2 cell intrinsic hypo-responsiveness is dependent upon the anergy factor Grail, suggesting similarities with adaptive tolerance [16].

In this study we identify the transcriptional changes associated with the development of *L. sigmodontis*-elicited Th2 cell-intrinsic hypo-responsiveness, and test whether it represents a unique and stable state of Th2 cell differentiation. We find that hypo-responsive Th2 cells retained their dysfunctional phenotype upon transfer to a naïve recipient indicating that their unresponsive state is stably maintained in the absence of antigen and active infection. We demonstrate that hypo-responsive Th2 cells have a divergent transcriptional profile to classical Th2 cells isolated either prior to the onset of hypo-responsiveness, or from mice acutely infected with the nematode *Nippostrongylus brasiliensis*. Although sharing characteristics with anergy and tolerance, hypo-responsive Th2 cells had a global transcriptional profile that differ-entiated them from exhausted, anergic, and tolerised CD4$^+$ T cells. Contrasting with their loss of ability to produce Th2 cytokines, hypo-responsive Th2 cells gained IL-21 production, and IL-21R blockade increased resistance to *L. sigmodontis* infection, indicating a novel regulatory role for IL-21. This indicates that Th2 cell-intrinsic hypo-responsiveness represents a distinct and stable state of T cell differentiation, and that hypo-responsive Th2 cells may inhibit Type 2 immunity via IL-21.

## Materials and methods

### Ethics statement

All animal work was approved by the University of Edinburgh Ethics Committee (PL02-10) and by the UK Home Office (PPL70/8548), and conducted in accordance with the Animals (Scientific Procedures) Act 1986.

### Animals, parasites, and cell isolations

Female BALB/c and IL-4gfp 4get reporter mice on the BALB/c background were bred in-house and maintained under specific pathogen-free conditions at the University of Edinburgh. Mice were used at 6–12 weeks of age, and randomly assigned to experimental groups. The *L. sigmodontis* life cycle was maintained in gerbils (*Meriones unguiculatus*) using the mite vector *Ornithonyssus bacoti* [22]. Mice were infected s.c. on the upper back with 30 *L. sigmodontis* L3 larvae. Adult or larval parasites were recovered by lavage of the thoracic cavity. *L. sigmodontis* antigen (LsAg) was prepared by collecting the PBS-soluble fraction of homogenized adult male

and female worms. To quantify blood microfilariae, 30 µL of tail blood was collected in FACS lysing solution (Becton-Dickinson), and microfilaria counted using a dark field optical microscopy (Axiovert 25, Zeiss). *N. brasiliensis* was maintained in Sprague-Dawley rats as previously described [23]. Mice were infected by s.c. injection with 200 *N. brasiliensis* L3 larvae. The parathymic, posterior, mediastinal and paravertebral LN, were taken as a source of thoracic LN (tLN) draining the pleural cavity (PleC). PleC cells were recovered by lavage. TLN cells were dissociated and washed in RPMI-1640 (invitrogen) supplemented with 0.5% mouse sera (Caltag-Medsystems), 100 U/ml penicillin, 100 µg/ml streptomycin and 2 mM L-glutamine.

## Flow cytometry and intracellular cytokine staining

The following antibodies were used: Pacific Blue- or Alexa 700-conjugated anti-CD4 (RM4-5), polyclonal anti-GFP (Ebioscience), Alexafluor488-conjugated goat anti-rabbit IgG (Invitrogen), phycoerythrin (Pe)-conjugated anti-IL-4 (11B11, Biolegend), allophycocyanine (APC)-conjugated anti-IL-5 (TRFK5, Biolegend), biotinylated anti-CXCR5 (RF8B2, BD Biosciences), PcPCy5.5-conjugated anti-CD44 (IM7, Biolegend), Pacific Blue-conjugated anti-CD62L (MEL-14, Biolegend), PeCy7-conjugated anti-CD25 (PC61, Biolegend), Pe-conjugated anti-CD45RB (C363-16A, Biolegend), Fitc-conjugated anti-IFNg (XMG1.2, Biolegend), Pe-conjugated anti-T-bet (4B10 Ebioscience), Pe-conjugated anti-Blimp1 (5E7, Biolegend), Pe-conjugated anti-c-Maf (sym0F1, Ebioscience), Pe-conjugated anti-Egr2 (erongr2, Ebioscience), Pe-conjugated anti-gp49 (H1.1, Biolegend), Alexa Fluor 647-conjugated anti-Granzyme B (GB11, Biolegend), Pe-conjugated anti-IL-21 (FFA21, Ebioscience), Fitc-conjugated anti-Gata3 (REA174, Miltenyi) and APC-conjugated Streptavidin (Biolegend). Non-specific binding was blocked with 4 µg of rat IgG/ 1x106 cells. Staining of transcription factors was performed using the Foxp3/Transcription Factor Staining Buffer Set (Ebioscience). For intracellular cytokine staining $2x10^6$ cells/well were stimulated in in RPMI-1640 media (Invitrogen), containing 100 U/mL Penicillin, 100 µg/ mL Streptomycin, 2 mM L-Glutamine, 0.5 µg/mL PMA and 1 µg/mL Ionomycin (both Sigma) at 37˚C and 5% $CO_2$ for 4 hours. Brefeldin-A (Invitrogen) was added at a final concentration of 10 µg/mL for the final 2 hours. Dead cells were excluded by the Zombie Aqua Fixable Viability kit (Biolegend), and the cells fixed and permeabilized using Biolegend Fixation Buffer in combination with the Intracellular Permeabilisation Wash Buffer. Flowcytometric acquisition was performed by using a FACS Canto II (BD Biosciences), data was analysed using Flowjo Software (Tree Star).

## *L. sigmodontis* specific antibody ELISA

ELISA plates (NUNC) were coated with 5 µg/ml LsAg diluted in 0.45M $NaHCO_3$/0.18M $Na_2CO_3$ (Sigma-Aldrich). Plates were incubated with serial dilutions of serum, and a representative dilution from the linear section of the dilution curve selected for each isotype (1/3200 for IgM and IgG2a, 1/6400 for IgG1). Detection of Ab isotypes was performed using HRP-conjugated anti-mouse IgG1, IgG2a or IgM (Southern Biotechnology Associates) and ABTS peroxidase substrate system (KPL).

## Adoptive transfers

CD4 T cells were purified by magnetic positive selection using anti-CD4 micro beads (Miltenyi Biotec). Sorted cells were > 93% positive for CD4. Due to limiting cell numbers, tLN and PleC cells were pooled from 12–18 infected mice for transfer into 9 recipients. The proportion of CD4+IL-4gfp+ Th2 cells in the day (d) 20 and d 60 samples were quantified by flow cytometry, and samples adjusted to transfer $4.3x10^5$ d 20 or d 60 CD4+IL-4gfp+ cells i.v. into wild-type BALB/c recipients. Recipients were challenged with *L. sigmodontis* 2-weeks post-transfer, and

autopsies performed 3-weeks post-transfer. Parasite numbers were quantified to confirm infections.

## RNA preparation and microarray

CD4 T cells were enriched from the PleC of *L. sigmodontis* infected mice, tLN of *L. sigmodontis* and *N. brasiliensis* infected mice, and spleen of naïve mice by magnetic negative selection (Dyna-Mag) using anti-CD8 (53–6.72), anti-CD19 (ID3), anti-MHC class II (M5/114.15.2), and anti-Mac1 (M1/70) followed by sheep anti-rat beads (Invitrogen). CD4$^+$IL-4gfp$^+$ and CD4$^+$IL-4gfp$^+$CXCR5$^-$ Th2 cells were then purified from the enriched PleC and tLN CD4$^+$ populations using a FACSAria flow sorter (BD Biosciences). Naïve splenic T cells were sorted based on CD4$^+$IL-4gfp$^-$CXCR5$^-$CD25$^-$CD44$^-$CD62L$^+$CD45RB$^+$. Due to limiting cell numbers, each biological replicate was created by pooling cells from 20–31 mice for day 20 *L. sigmodontis* infection, 16–21 mice for day 60 *L. sigmodontis* infection, 20–23 for *N. brasiliensis* infection, and 3 naïve mice.

Purities of sorted cells were: naïve spleen (97–99%), *N. brasiliensis* tLN (94–97%), *L. sigmodontis* d 20 tLN (87–96%), *L. sigmodontis* d 60 tLN (94–98%), *L. sigmodontis* d 20 PleC (91–98%), and *L. sigmodontis* d 60 PleC (96–99.5%). Biological replicates were created using independent infections. Purified cells were shock-frozen and stored at -80˚C until RNA purification using the RNeasy Kit (Qiagen) and RNA quantitated by Qubit (ThermoFisher Scientific). Samples were processed and hybridised to an Affymetrix 2.1 ST mouse array by Edinburgh Genomics. All subsequent array data processing and analyses were undertaken in the R environment using Bioconductor packages. Array quality was assessed using the arrayQuality-Metrics package in Bioconductor to identify outliers [24]. One replicate failed the hybridisation process and 2 replicates failed quality control resulting in 5 independent biological replicates for naïve spleen, *N. brasiliensis* tLN, d 20 *L. sigmodontis* tLN, and d 60 *L. sigmodontis* tLN, 4 independent replicates for d 20 *L. sigmodontis* PleC, and 3 independent replicates for d 60 *L. sigmodontis* PleC.

## Microarray data analysis

Data were normalized using the robust multi-array average expression measure [25]. Pairwise group comparisons, using the linear modelling methods, with subsequent Bayesian approaches, provided in the limma Bioconductor package, were used to assess differential gene expression, with p-value adjustment for multiple testing. Adjusted $p<0.05$ was used to define significance for all gene array analysis. Fold-changes (log2-based) between groups, with filtering by adjusted (adj) P value (adj. $p<0.05$) as appropriate, were plotted using the heat-map.2 function. Principal component analyses were undertaken separately for all normalised and statistically significant data using the prcomp function prior to plotting using scatter-plot3d. Correlations were performed on log2 normalised sample data using Standard Pearson correlation. Gene array data is accessible at the Gene Expression Omnibus with accession code GSE114308. To identify genes expressed by exhausted T cells during LCMV infection [26] we downloaded the GEO GSE41870 dataset, and identified genes that significantly changed (adj. $p < 0.05$) between effector (D8, Armstrong) and exhausted (D30, Clone 13) CD4$^+$ T cells. To identify genes expressed by anergic T cells [27] we used the GEO GSE46242 dataset, and identified genes that significantly changed (adj. $p < 0.05$) between control EV and Anergic EV samples.

## In vivo IL-21R blockade

Mice received i.p. injections of 500 µg anti-IL-21R antibody (4A9, Bioxcell), or an equivalent dose of rat IgG (Sigma-Aldrich), on d 45, 48, 52, and 55 pi.

## Statistics for non-microarray data

Power calculations for group sizes were performed using G*power. Statistical analysis was performed using JMP version 12 (SAS). Parametric analysis of combined data from multiple repeat experiments, or of experiments containing more than two groups, was performed using ANOVA followed by Tukey's post-hoc tests when required. Adult parasite numbers were analysed using a GLM with a poisson distribution. When using two-way ANOVA or GLM to combine data from multiple experiments, experimental effects were controlled for in the analysis and it was verified that there were no significant qualitative interactions between experimental and treatment effects.

# Results

## Intrinsically hypo-responsive Th2 cells retain their dysfunctional phenotype in the absence of antigen and active infection

Filarial parasites are highly immunosuppressive and it is not known whether Th2 cell-intrinsic hypo-responsiveness is actively maintained by the parasite. The dependence of hypo-responsive Th2 cells on antigen is also unknown. Using IL-4gfp 4get reporter mice to track Th2 cells, we previously demonstrated that IL-4gfp$^+$ Th2 cells become functionally hypo-responsive between days (d) 20 and 60 of *L. sigmodontis* infection, showing a progressive loss in their intrinsic ability to proliferate and produce Th2 cytokines [19,20,28]. Thus, day 20 pi represents a time point before the onset of hypo-responsiveness when Th2 cells are still functionally active, whilst at d 60 pi Th2 cells are intrinsically hypo-responsive [19].

To test whether the maintenance of *L. sigmodontis*-induced Th2 cell-intrinsic hypo-responsiveness requires the presence of antigen or active infection, or is stably maintained in their absence, we purified CD4$^+$ T cells from *L. sigmodontis* infected BALB/c IL-4gfp mice at d 20 and 60 post-infection (pi) and adoptively transferred them into WT recipients. IL-4gfp mice are a valuable tool for tracking Th2 cells because they report IL-4 mRNA independently of IL-4 protein, and retain GFP expression for at least 4-weeks after clearance of infection [19,29]. Whole CD4$^+$ T cells were pooled from the infection site, the pleural cavity (PleC), and the LN within the thoracic cavity (tLN) that drain the PleC [19,20], as Th2 cell-intrinsic hypo-responsiveness occurs at both locations and cell numbers were limiting. GFP expression was used as a surrogate marker allowing us to specifically track and functionally assess the transferred *L. sigmodontis*-elicited IL-4 mRNA$^+$ Th2 cells. The total number of transferred CD4$^+$ T cells was adjusted to ensure transfer of equal numbers of d 20 and d 60 CD4$^+$IL-4gfp$^+$ T cells. Recipients were challenged with *L. sigmodontis* two-weeks post-transfer or left unchallenged. The survival and function of IL-4gfp$^+$ Th2 cells recovered from the PleC, brachial LN draining the skin inoculation site, and tLN, were assessed at d 7 post-challenge (Fig 1A).

In the absence of *L. sigmodontis* challenge, transferred d 20 and d 60 CD4$^+$IL-4gfp$^+$ T cells were detectable in the brachial LN, tLN, and PleC, and showed equivalent survival after three weeks (Fig 1B). Similarly, they showed equivalent expression of memory markers, with the majority of transferred IL-4gfp$^+$ Th2 cells displaying a CD62L$^-$CD44$^{hi}$ effector memory phenotype or CD62L$^+$CD44$^{hi}$ activated/central memory phenotype, independently of whether or not they received challenge infection (Fig 1C).

Upon challenge with *L. sigmodontis*, functional differences became apparent, and d 60 CD4$^+$IL-4gfp$^+$ Th2 cells showed a significantly reduced ability to expand (Fig 1B), and produce IL-4 and IL-5 protein (Fig 1D & 1E). Thus, hypo-responsive d 60 IL-4gfp$^+$ Th2 cells showed equivalent survival and expression of memory markers compared to active d 20 Th2 cells upon transfer to a naïve recipient. However, they retained their dysfunctional phenotype,

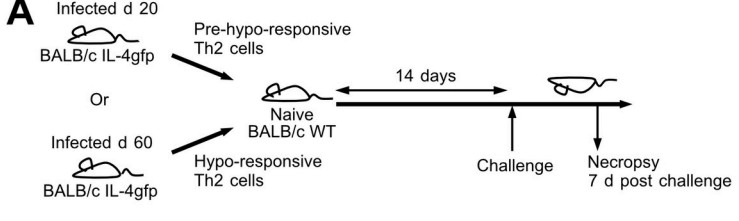

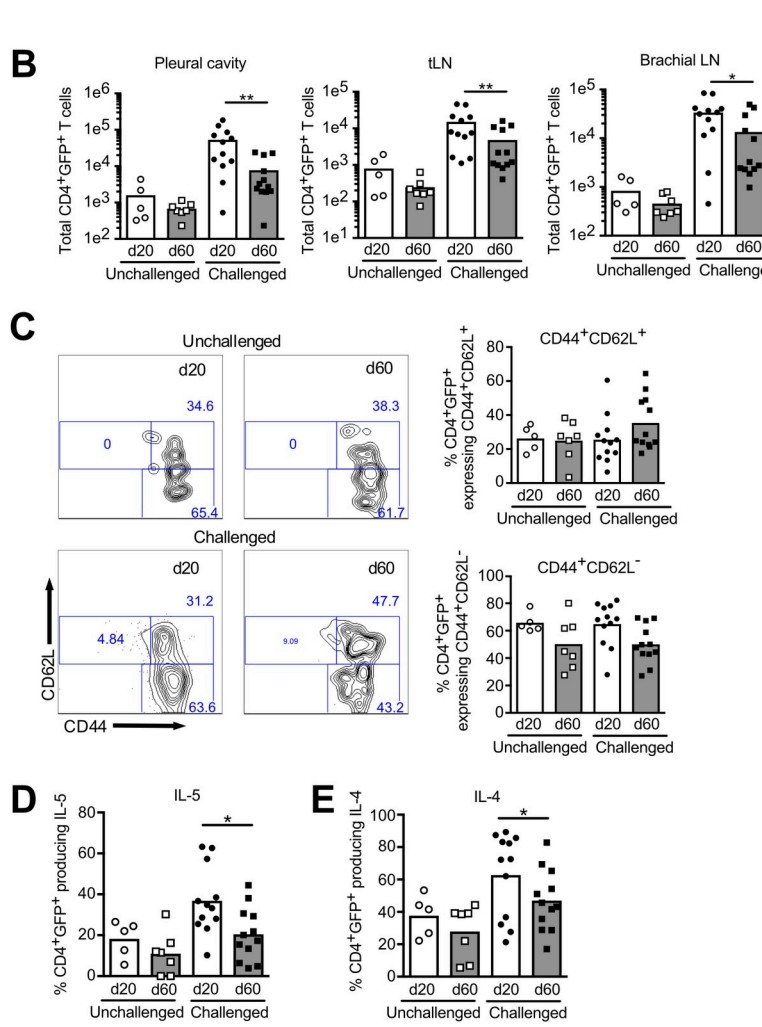

**Fig 1. Hypo-responsive Th2 cells retain their dysfunctional phenotype in the absence of antigen.** CD4[+] T cells from *L. sigmodontis* infected IL-4gfp mice were isolated at d 20 (circles) and d 60 (squares) pi and transferred into naive WT recipients. WT recipients were either left unchallenged (open symbols), or challenged (closed symbols) with *L. sigmodontis* 2-weeks post-transfer, and IL-4gfp[+] Th2 cell function assessed 3 weeks post-transfer. Figures show data from two independent experiments (n = 5–6 for unchallenged, n = 12 for challenged). Symbols represent individual animals, and bars show mean values. (A) Schematic of experimental approach. (B) Total numbers of CD4[+]IL-4gfp[+] donor cells within the PleC, tLN, and brachical LN of recipient. $^*p<0.05$, $^{**}p<0.01$ (ANOVA based on combined data from two independent experiments). (C) Flow plots show representative expression of CD62L and CD44 by donor CD4[+]IL-4gfp[+] Th2 cells recovered from the PleC. Figures show proportions of CD44[+]CD62L[+] and CD4[+]CD62L[-] CD4[+]IL-4gfp[+] Th2 cells. (D) Proportion of CD4[+]IL-4gfp[+] PleC Th2 cells producing IL-5. $^{**}$Significant effect of timepoint, $p<0.05$ (ANOVA based on combined data from two independent experiments). (E) Proportion of CD4[+]IL-4gfp[+] PleC Th2 cells producing IL-4. $^*$Significant effect of timepoint, $p<0.05$ (ANOVA based on combined data from two independent experiments).

confirming an intrinsic functional change and that this phenotype is maintained for at least 2 weeks in the absence of antigen and active infection.

## Intrinsically hypo-responsive Th2 cells and classically active Th2 cells have divergent transcriptional profiles

To investigate whether Th2 cell-intrinsic hypo-responsiveness associates with the development of a unique gene expression profile, an Affymetrix 2.1 mouse gene ST array platform was used to compare mRNA expression of CD4+IL-4gfp+ Th2 cells purified from the PleC and tLN of *L. sigmodontis*-infected mice pre- (d 20) and post- (d 60) onset of hypo-responsiveness (Fig 2A & 2B, S1A Fig). TLN Th2 cells were also defined as being CXCR5- to exclude CXCR5+ T follicular helper (Tfh) cells, which are IL-4gfp+ during helminth infection [30,31]. To act as a reference effector Th2 cell population, CD4+IL-4gfp+CXCR5- Th2 cells were purified from the tLN of mice infected with *Nippostrongylus brasiliensis* (S1B Fig). *N. brasiliensis* is cleared within 7–9 days, stimulates a prototypical acute Th2 response, and the tLN drain the lung migration stage of infection. To identify the gene expression profile of naïve T cells, CD4+IL-4gfp-CXCR5-CD25-CD44-CD62L+CD45RB+ T cells were purified from the spleens of uninfected mice (S1C Fig).

To determine the relatedness of the individual samples to each other we performed unsupervised clustering using hierarchical analysis (Euclidean distance measure) with normalised gene expression levels as input. The individual samples clustered back into their original groups demonstrating the reliability of the gene expression data, and that the transcriptional differences between groups are greater than between biological replicates (Fig 2C). As expected, naïve CD4+ T cells had a distinct mRNA expression profile to the IL-4gfp+ Th2 cell populations and clustered separately. This was confirmed by principal component analysis, which also showed that *N. brasiliensis* Th2 cells clustered separately from *L. sigmodontis* Th2 cell populations indicating infection-specific transcriptional changes (Fig 2D). IL-4gfp+ Th2 cells from the tLN and PleC of *L. sigmodontis* infected mice also showed different transcriptional profiles indicating location specific gene expression (Fig 2C & 2D). Importantly, the transcriptional profiles of *L. sigmodontis* IL-4gfp+ Th2 cells differed over time, consistent with their phenotype change. In particular, the hypo-responsive d 60 PleC IL-4gfp+ Th2 cells segregated separately from all the other Th2 cell populations, including the d 20 PleC IL-4gfp+ Th2 cells (Fig 2C & 2D).

Th2 cells from different contexts show both common and specific gene expression profiles [32]. Similarly, based on PleC IL-4gfp+ Th2 cells, 4124 common loci significantly changed upon infection with *L. sigmodontis* and *N. brasiliensis*, whilst 4155 and 2308 unique loci significantly changed upon *L. sigmodontis* and *N. brasiliensis* infections respectively (Fig 2E). A similar pattern was seen for tLN IL-4gfp+ Th2 cells, although there were fewer *L. sigmodontis* unique loci. Only 51% of loci that significantly changed in PleC IL-4gfp+ Th2 cells at d 60 of *L. sigmodontis* infection were common with those that changed at d 20 pi.

These transcriptional changes over time during *L. sigmodontis* infection associated with the differential expression of transcription factors. Investigating transcription factors that had significantly differential gene expression compared with naïve T cells demonstrated that d 20 and d 60 IL-4gfp+ Th2 cells had unique expression of 43 and 138 transcription factors respectively (S2 Fig), and had 152 commonly expressed transcription factors (S3 Fig). Day 60 IL-4gfp+ Th2 cells also showed enrichment of particular KEGG pathways compared with d 20 IL-4gfp+ Th2 cells (S1 Table). In particular, d 60 PleC IL-4gfp+ Th2 cells showed enrichment of metabolism pathway genes, and both d 60 PleC and tLN IL-4gfp+ Th2 cells showed distinct enrichment of DNA repair pathway genes. Altogether this indicates that intrinsically hypo-responsive d 60

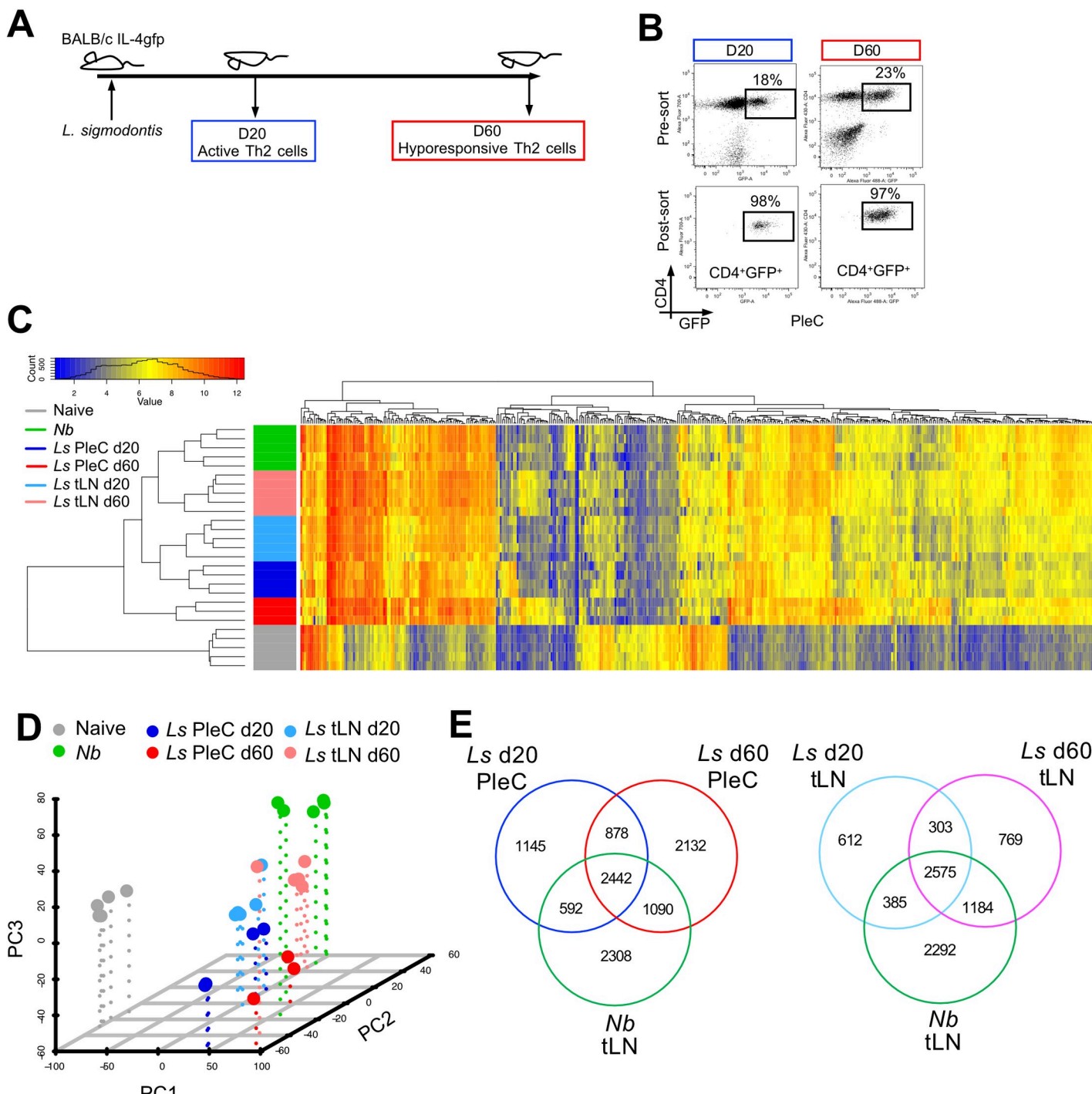

**Fig 2. Hypo-responsive Th2 cells have a distinct gene expression profile.** (A) Schematic showing time points at which active and hypo-responsive IL-4gfp+ Th2 cells were purified from *L. sigmodontis* infected mice. (B) Representative flow plots showing pre- and post- sorts for CD4+IL-4gfp+ Th2 cells from the PleC of *L. sigmodontis* infected mice at d 20 and d 60 pi. (C) Unbiased hierarchical clustering of IL-4gfp+ Th2 cells purified from the PleC and tLN of mice infected with *L. sigmondontis* (*Ls*) and from the tLN of *N. brasiliensis* (*Nb*) infected animals d 6 pi., and naïve splenic IL-4gfp-CD4+ T cells. Clustering was performed based on Euclidean distance using the fold change of loci that exhibited an 8-fold or greater change in one or more comparisons. (D) Principal component analysis based on all normalised data. Symbols represent individual samples. (E) Venn diagrams showing the number of loci from *L. sigmodontis* (*Ls*) and *N. brasiliensis* (*Nb*) Th2 cells that significantly differ compared to naïve (adjusted p<0.05).

IL-4gfp$^+$ Th2 cells are transcriptionally divergent from classically active Th2 cells from both d 20 of *L. sigmodontis* infection and from *N. brasiliensis* infection.

## Hypo-responsive Th2 cells retained a Th2 transcriptional profile, while a subset upregulated T-bet and produced IFN-γ

The loss of Th2 cytokines by hypo-responsive IL-4gfp$^+$ Th2 cells could indicate a switch in phenotype towards a different T cell subset rather than a manifestation of Th2 cell dysfunction. To address this, the expression of the major transcription factors for Th2 (*Gata3*, *Irf4*), Th1 (*Tbx21*), Th9 (*Spi1*), Th17 (*Rorc*), Treg (*Foxp3*), Tr1 (*Irf1*), and Tfh (*Bcl6*) cells were assessed. All IL-4gfp$^+$ Th2 cells from *L. sigmodontis* and *N. brasiliensis* infected mice showed significant upregulation of the Th2 transcription factors *Gata3* and *Irf4* compared to naïve T cells, and their expression remained constant between d 20 and d 60 of *L. sigmodontis* infection (Fig 3A). *Bcl6* was significantly increased in CD4$^+$IL-4gfp$^+$CXCR5$^-$ tLN T cell populations from both *L. sigmodontis* and *N. brasiliensis* infected mice, suggesting they may contain IL-4gfp$^+$Bcl6$^+$CXCR5$^-$ Tfh cells. Importantly, neither d 20 or d 60 IL-4gfp$^+$ Th2 cells from the PleC, where Th2 cell-intrinsic hypo-responsiveness predominates, showed elevated *Bcl6* mRNA compared with naïve T cells. Thus, the hypo-responsive PleC d 60 Th2 population did not take on the characteristics of Tfh cells.

Expression of mRNA for *Spi1* and *Rorc* were not significantly upregulated in any of the IL-4gfp$^+$ T cell populations relative to naïve T cells. *Irf1* mRNA levels were significantly reduced in IL-4gfp$^+$ Th2 cells compared to naïve T cells, although there was a significant increase between d 20 and 60 of *L. sigmodontis* infection. Interestingly, despite impaired production of Th2 cytokines and IL-10 at the protein level [19,20], expression of *Il5* mRNA remained constant between d 20 and 60, and expression of *Il10*, *Il13*, and *Il4* mRNA significantly increased in PleC IL-4gfp$^+$ Th2 cells (Fig 3B). There was no significant upregulation of mRNA for IL-9 or IL-17 during *L. sigmodontis* infection. Thus, the dysfunctional d 60 IL-4gfp$^+$ T cell population does not take on characteristics of Tfh, Th9, Th17, Tr1, or Foxp3$^+$ Treg populations, and instead retains a typical Th2 cell mRNA phenotype with the inhibition of Th2 cytokines occurring post-transcriptionally.

*Tbx21* (T-bet) was significantly upregulated between d 20 and 60 of *L. sigmodontis* infection (Fig 3A), and *Ifng* was the most upregulated loci (27-fold, adjusted p = 4.1e$^{-6}$) between PleC d 60 and d 20 IL-4gfp$^+$ Th2 cells (Fig 3B). Consistent with the development of a Th1 phenotype, Ingenuity Pathway Analysis (IPA) predicted IFN-γ (p = 2.7e$^{-24}$, F = 4.1) (S4 Fig), IL-12 (complex) (p = 2.7e$^{-24}$, F = 3.9), IL-12A (p = 6.93e$^{-11}$, F = 2.6), IL-12B (p = 6.93e$^{-11}$, F = 2.6), and IL-18 (p = 3.17e$^{-11}$, F = 3.2) to be activated upstream regulators, explaining some of the observed gene expression changes between d 60 and d 20 PleC IL-4gfp$^+$ Th2 cells.

Flow cytometry was used to test whether cells with a mixed Th1/Th2 phenotype develop during *L. sigmodontis* infection, as described for other helminth infections [33]. In naïve mice, just under 40% of PleC IL-4gfp$^+$ Th2 cells produced IFN-γ protein (Fig 3C–3E), with the majority of IFN-γ$^+$ cells co-producing IL-4 (Fig 3C & 3D). Upon *L. sigmodontis* infection, the proportion of PleC IL-4gfp$^+$IFN-γ$^+$ Th2 cells significantly reduced to less than 5% at d 20 consistent with a dominant Th2 response at this time point, before significantly increasing during the hypo-responsive stage at d 60 pi (Fig 3C–3E). Although co-staining for T-bet and IL-4gfp was not possible, expression of T-bet by CD4$^+$ T cells significantly increased between d 20 and 60 pi (Fig 3F). The IFN-γ production by d 60 IL-4gfp$^+$ Th2 cells was retained following transfer to a naïve recipient and rechallenge with *L. sigmodontis* (Figs 1A & 3G), indicating that the hypo-responsive IL4gfp$^+$ Th2 cells also stably retained a Th1 phenotype.

During *L. sigmodontis* infection the d 60 IL-4gfp$^+$ Th2 cells that produced IFN-γ were distinct from those producing IL-4 and IL-5 (Fig 3C & 3D), contrasting with the published Th1/

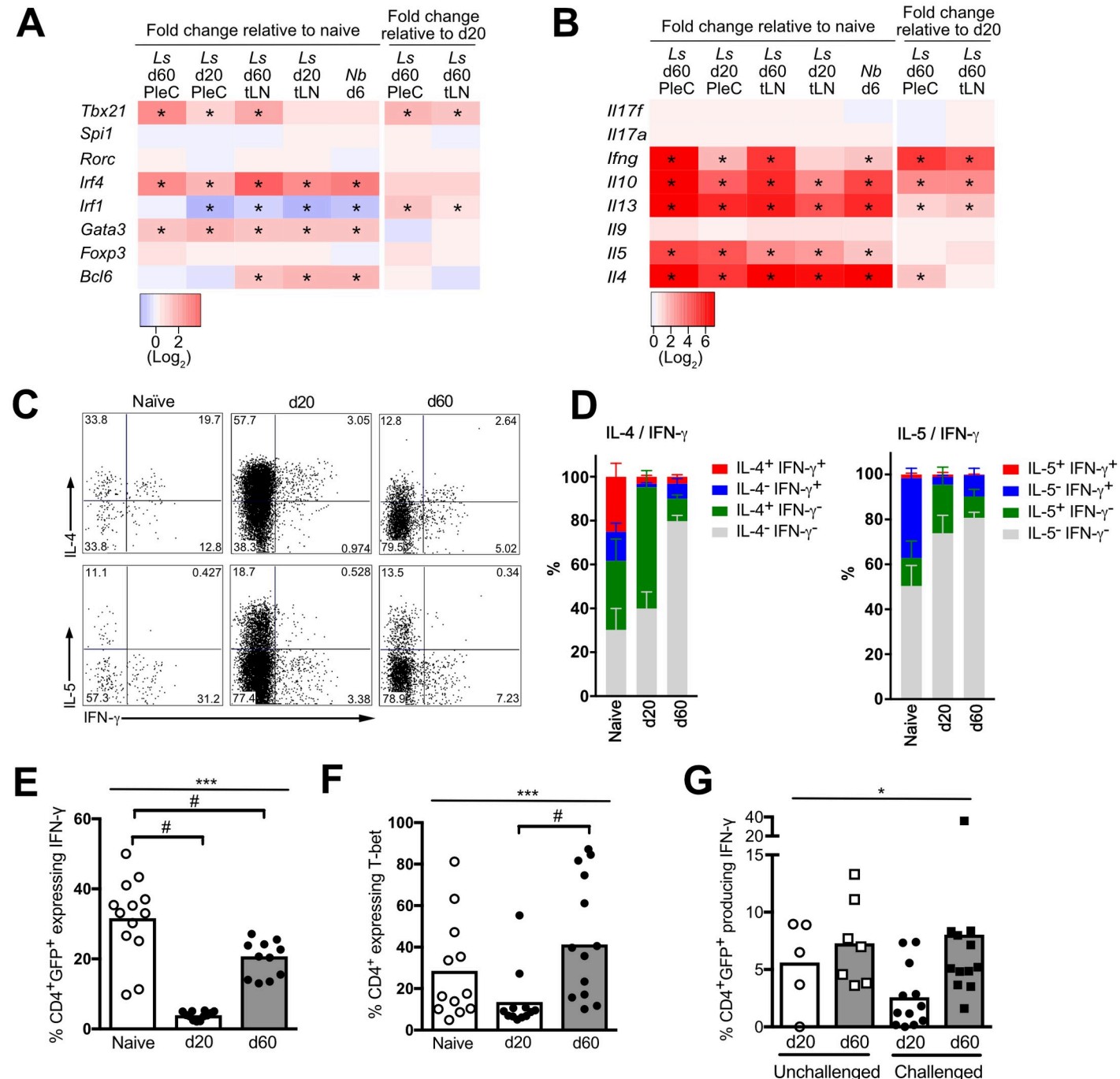

**Fig 3. A subset of d 60 IL-4gfp⁺ Th2 cells take on Th1 characteristics.** (A & B) Heatmaps showing fold change in expression of transcription factors (A) and cytokines (B) in *L. sigmodontis* (*Ls*) and *N. brasiliensis* (*Nb*) IL-4gfp⁺ cells relative to naïve IL-4gfp⁻CD4⁺ T cells, or in d 60 *L. sigmodontis* IL-4gfp⁺ Th2 cells relative to d 20. *adj. p<0.05). (C) Representative staining for production of IL-4, IL-5 and IFN-γ by PleC IL-4gfp⁺ from naïve and *L. sigmodontis* infected mice (d 20 and d 60 pi) (D) Proportion of IL-4gfp⁺ Th2 cells from the PleC of naïve and *L. sigmodontis* infected mice singly or co-expressing IL-4 and IFN-γ, or IL-5 and IFN-γ. Data shown from one representative experiment of two (n = 3 for naïve and 6 for *L. sigmodontis* infected mice). (E) Proportion of IL-4gfp⁺ Th2 cells from the PleC of naïve (open symbols) or d 20 and d 60 *L. sigmodontis* infected mice (closed symbols) producing IFN-γ. Symbols show individual mice and bars represent mean values. Graph shows combined data from two independent experiments. ***Significant effect between groups (p<0.001, ANOVA), # p<0.05 (Tukey's HSD). (F) Proportion of PleC CD4⁺ T cells from naïve and *L. sigmodontis* infected mice expressing T-bet. Symbols show individual mice and bars represent mean values. Graph shows combined data from two independent experiments. ***Significant effect between groups (p<0.001, ANOVA), # p<0.05 (Tukey's HSD). (G) CD4⁺ T cells from *L. sigmodontis* infected IL-4gfp mice were isolated at d 20 (circles) and d 60 (squares) pi and transferred into naïve WT recipients. Recipients were either left unchallenged (open symbols), or challenged (closed symbols) with *L. sigmodontis* two weeks post-transfer, and the proportion of PleC IL-4gfp⁺ Th2 cells producing IFN-γ assessed 3 weeks post-transfer. *Significant effect of time point independent of challenge status (p<0.05, ANOVA based on combined data from two independent experiments).

Th2 hybrid phenotype [33]. Importantly, the loss of IL-4 (15.8%) and IL-5 (16.4%) producing Th2 cells was greater than the gain in IFN-γ producing Th2 cells (5.5%) (Fig 3D), indicating that Th2 cell intrinsic hypo-responsiveness does not represent a switch to a Th1 phenotype. This suggests that a proportion of the hypo-responsive d 60 IL-4gfp+ Th2 population took on Th1 characteristics. However, as a whole, the d 60 IL-4gfp+ Th2 cells retained a Th2 phenotype and a Th2 to Th1 switch was not responsible for their dysfunctional phenotype.

### Intrinsically hypo-responsive CD4+ Th2 cells express Blimp-1 and c-Maf, but have a distinct transcriptional profile to exhausted CD4+ T cells

PD-1 co-inhibition is a key mechanism of T cell exhaustion [34], suggesting that PD-1-dependent Th2 cell-intrinsic hypo-responsiveness represents a form of exhaustion. To investigate this hypothesis, we compared the gene expression profiles of hypo-responsive Th2 cells and exhausted CD4+ T cells. Published CD4+ T cell gene expression data from LCMV infection was used as a well-defined exhaustion model [26]. We initially assessed whether intrinsically hypo-responsive Th2 cells and exhausted CD4+ T cells upregulate the same transcription factors. Of the 17 exhaustion-associated transcription factors, only 5 (*Maf*, *Mcm6*, *Baz1a*, *Mdfic*, and *Prdm1*) were significantly upregulated in PleC IL-4gfp+ Th2 cells between d 20 and 60 pi, and all five showed similar expression levels in our reference active *N. brasiliensis* IL-4gfp+ Th2 cells (Fig 4A). Expression of Blimp1 and c-Maf protein did correlate with Th2 cell hypo-responsiveness as they were highly expressed in CD4+ T cells at d 60, but not d 20, pi (Fig 4B & 4C), and IPA predicted that c-Maf was an activated upstream regulator of PleC d 60 IL-4gfp+ Th2 cells (z-score = 2.2, p = 6e$^{-6}$). However, d 60 IL-4gfp+ Th2 cells showed an inverse gene expression pattern to exhausted CD4+ T cells (r = -0.170, p<0.001), based on genes that are significantly differentially expressed in exhausted CD4+ T cells [26] (Fig 4D). Thus, hyporesponsive Th2 cells and exhausted T cells have common expression of Blimp1 and c-Maf, but overall show inverse gene expression profiles indicating that they represent distinct T cell activation states.

### Th2 cell-intrinsic hypo-responsiveness has similarities with T cell anergy and tolerance

Immune suppression during filariasis and other helminth infections is associated with increases in T cell anergy markers, including upregulation of the E3 ubiquitin ligases Nedd4, Itch, C-cbl, Cbl-b, and GRAIL [13–16]. To test whether Th2 cell-intrinsic hypo-responsiveness represents a form of anergy, the expression of anergy-associated E3 ubiquitin ligases was assessed (Fig 5A). *Nedd4* and *Rnf128* (GRAIL) mRNA were significantly increased in PleC IL-4gfp+ Th2 cells at both d 20 and d 60 of *L. sigmodontis* infection in comparison to naïve CD4+ T cells, and were not significantly upregulated in *N. brasiliensis* IL-4gfp+ Th2 cells. However, their expression did not change significantly between d 20 and d 60 of *L. sigmodontis* infection. *Ndfip1* was significantly upregulated during *L. sigmodontis* infection, and its expression significantly increased between d 20 and d 60 (Fig 5A). However, it was also upregulated to a similar extent in in *N. brasiliensis* IL-4gfp+ Th2 cells. Expression of other anergy-associated ubiquitin ligases (*Rc3h1*, *Cbl*, *Cblb*, *Cblc*, *Itch*, and *Traf6*) either decreased upon *L. sigmodontis* infection or remained unchanged.

Egr2 is a key anergy associated transcription factor [27,35], and PleC IL-4gfp+ Th2 cells significantly increased expression of Egr2 protein between d 20 and d 60 pi (Fig 5B). Comparing the fold change of genes that significantly differed in CD4+ T cells following Egr2-dependent anergy induction [27] against their fold-change in IL-4gfp+ Th2 cells between d 20 and d 60 of *L. sigmodontis* infection showed a significant positive correlation (r = 0.376, p<0.001) (Fig

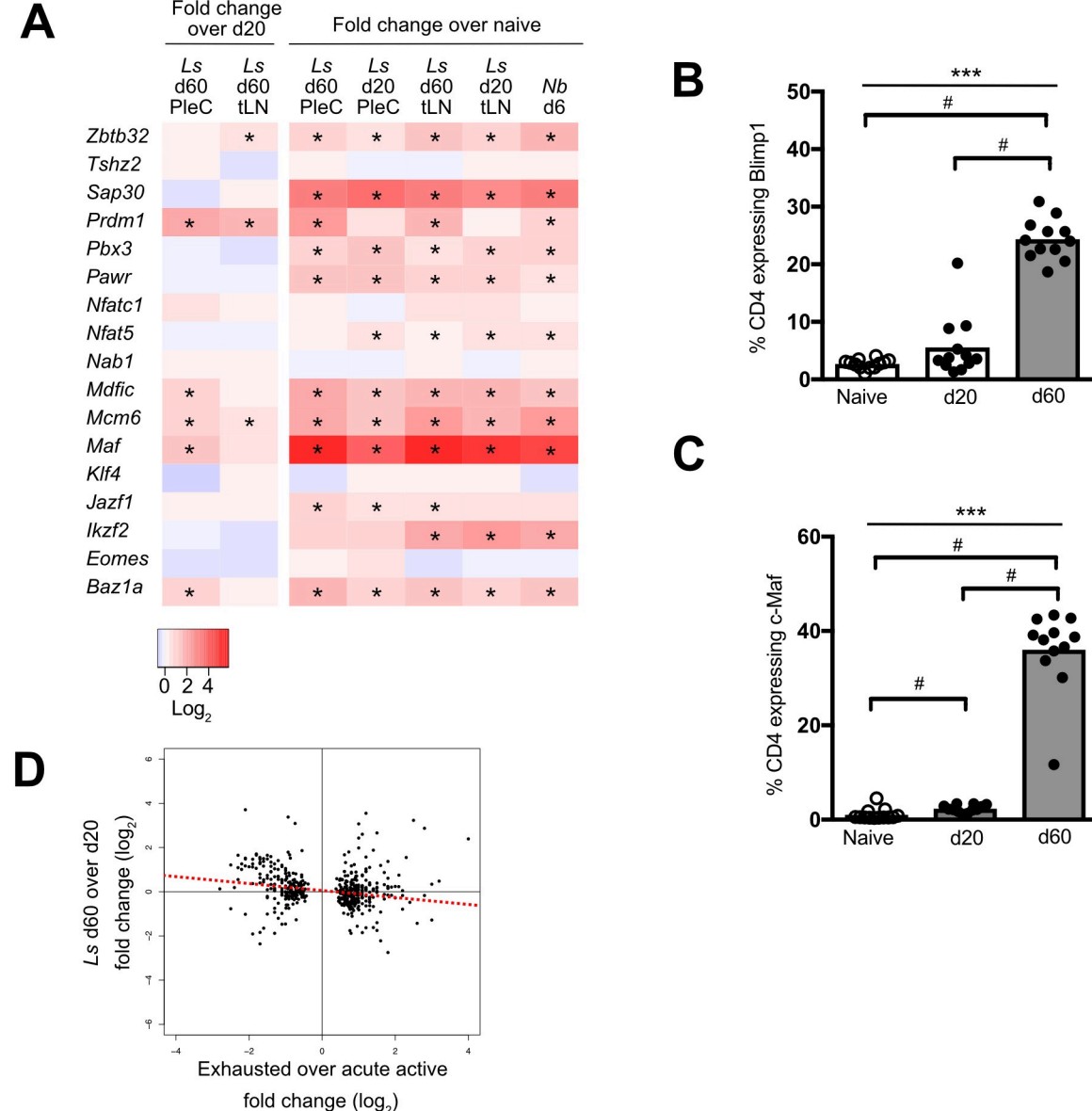

**Fig 4. Hypo-responsive Th2 cells are distinct from exhausted CD4+ T cells.** (A) Heatmap showing fold change in expression of exhaustion-associated transcription factors in *L. sigmodontis* (*Ls*) and *N. brasiliensis* (*Nb*) IL-4gfp+ cells relative to naïve IL-4gfp- CD4+ T cells, or in d 60 *L. sigmodontis* IL-4gfp+ Th2 cells relative to d 20. *adj. p<0.05. (B & C) Proportion of PleC CD4+ T cells expressing Blimp1 (B) and c-Maf (C) in naïve and *L. sigmodontis* infected mice at 20 and 60 d pi. Symbols represent individual animals, bars show mean. Naïve animals from d 20 and d 60 timepoints were pooled. Combined data from two experiments shown. ***Significant group effect (p<0.001, ANOVA using combined data from both experiments), # p<0.05 (Tukey HSD). (D) Fold-change of genes significantly differentially expressed (adj. p<0.05) in exhausted CD4+ T cells correlated against their fold-change expression in hypo-responsive Th2 cells (Pearson Rank Correlation).

5C). When compared with d 20, d 60 PleC IL-4gfp+ Th2 cells also significantly upregulated 9 out of the 10 key genes associated with the induction of T cell tolerance following peptide immunotherapy [36], including *Lag3*, *Tigit*, *Havcr2*, *Icos*, *Il10*, *Il21*, *Maf*, *Nfil3*, and *Ahr*. However, *L. sigmodontis* IL-4gfp+ Th2 cells only upregulated a subset of the total genes upregulated in Egr2-dependent anergy (Fig 5D, S2 Table) or peptide-induced tolerance (S5 Fig & S3 Table), and the d 60 IL-4gfp+ Th2 cells showed similar expression levels to the reference *N.*

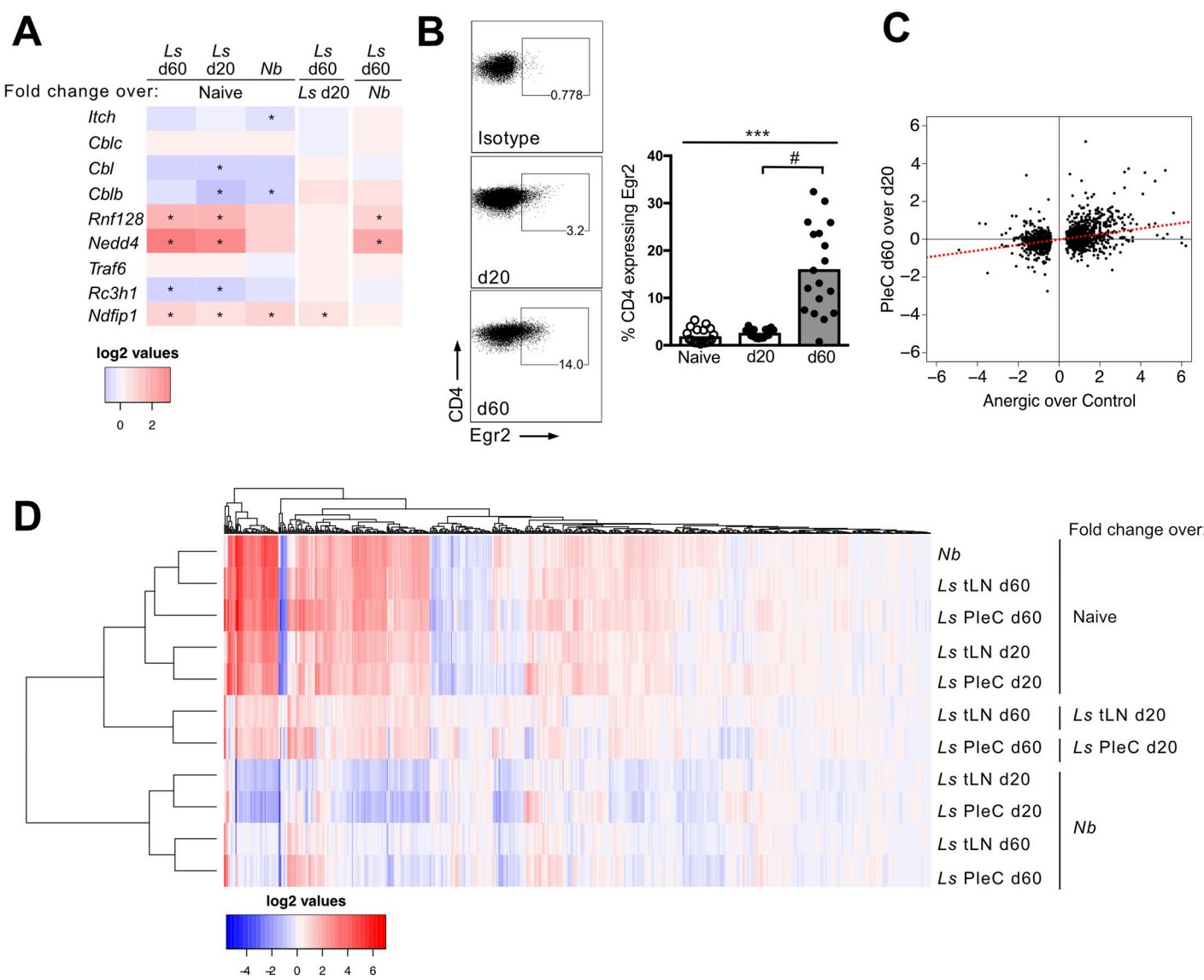

**Fig 5. Hypo-responsive Th2 cells have similarities with T cell anergy and tolerance.** (A) Heatmap showing fold change in expression of anergy-associated ubiquitin ligases in *L. sigmodontis* (*Ls*) and *N. brasiliensis* (*Nb*) IL-4gfp⁺ cells respectively, relative to naïve IL-4gfp⁻CD4⁺ T cells, and in d 60 *L. sigmodontis* IL-4gfp⁺ Th2 cells relative to d 20 and *N. brasiliensis* Th2 cells. Data is from PleC IL-4gfp⁺ Th2 cells for *L. sigmodontis* infection. *adjusted p<0.05. (B) Representative flow staining and proportion of CD4⁺ T cells expressing Egr2 in naïve and *L. sigmodontis* infected mice 20 and 60 d pi. Figure shows data from two independent experiments. Symbols represent individual animals and bars show means. *** Significant difference between groups (p<0.001, ANOVA using combined data from both experiments), # p<0.05 (Tukey HSD). (C) Fold-change of genes significantly up- or down- regulated in anergic CD4⁺ T cells correlated against their fold-change expression in hypo-responsive Th2 cells (Pearson Rank Correlation). (D) Unbiased hierarchical clustering of IL-4gfp⁺ Th2 cells purified from the PleC and tLN of mice infected with *L. sigmondontis* (*Ls*) and from the tLN of *N. brasiliensis* (*Nb*) infected animals. Clustering was performed based on Euclidean distance using the fold change of genes that are significantly (adj. p < 0.05) increased in anergic CD4⁺ T cells.

*brasiliensis* IL-4gfp⁺ Th2 cells. Together this indicates that IL-4gfp⁺ Th2 cells upregulate some anergy and tolerance associated genes during *L. sigmodontis* infection, and that d 60 IL-4gfp⁺ Th2 cells have developed a gene expression profile more similar to that of anergic or tolerised CD4⁺ T cells compared with d 20 IL-4gfp⁺ Th2 cells. However, despite a partial overlap with tolerised and anergic T cells, the gross transcriptional profile of Th2 cell-intrinsic hypo-responsiveness was dissimilar to T cell anergy and tolerance.

## Identification of immune regulatory genes associated with Th2 cell-intrinsic hypo-responsiveness

1132 and 257 loci significantly changed in PleC and tLN IL-4gfp$^+$ Th2 cells respectively between d 20 and d 60 of *L. sigmodontis* infection (S4 Table & S5 Table). We predicted that genes responsible for the dysfunctional Th2 cell phenotype should show a different expression profile to the reference *N. brasiliensis* Th2 cells. 471 of the PleC loci were also significantly differentially expressed compared with *N. brasiliensis* Th2 cells, and thus formed a core list of candidate loci (S6 Table). Th2 cell-intrinsic hypo-responsiveness is detected to a lesser extent in the tLN, so we predicted that candidate loci should show a similar expression pattern in the tLN, but to a lower level. Thus, the core candidate list was refined based on their expression in the tLN. 308 loci showed a similar expression profile in the tLN, of which 58 changed significantly (S7 Table). 18 loci (17 unique genes) matched our full prediction by having a similar expression pattern in the tLN and being expressed to significantly lower levels in the tLN compared to the PleC (Fig 6A).

As the gene array data indicated post-transcriptional inhibition of Th2 cytokines (Fig 3B) we investigated the expression of translation-associated genes based on KEGG pathways, IPA-defined 'translation regulators', and literature searches. Nine translation associated-genes were identified amongst the candidate genes (*Pwp2*, *Riok1*, *Gtpbp4*, *Lsg1*, *Mdn1*, *Utp20*, *Paip2b*, *Nup43*, *Ears2*), of which all but *Riok1* and *Nup43* were downregulated at d 60 compared with d 20. Interestingly, 28 eukaryotic type translation factors (KEGG Brite Ko03012) were significantly downregulated in d 60 PleC IL-4gfp$^+$ Th2 cells compared with naïve T cells, but only 7 of these were downregulated in *N. brasiliensis* Th2 cells (Fig 6B). Conversely, 10 translation factors were significantly upregulated in *N. brasiliensis* Th2 cells, of which only 2 were upregulated during *L. sigmodontis* infection. However, the expression of these 38 translation factors did not significantly differ between d 20 and d 60 of *L. sigmodontis* infection, indicating that their expression was specific to infection type rather than the change in Th2 cell phenotype over time. Nevertheless, *L. sigmodontis* infection associates with a reduction in expression of translation factors in Th2 cells when compared with acute *N. brasiliensis* infection.

Literature searches were used to identify and group candidate genes with known immune regulatory functions. In common with other forms of T cell dysfunction, various inhibitory receptors were upregulated (Fig 6C). Despite its central role in driving Th2 cell-intrinsic hypo-responsiveness [19], *Pd1* was not identified as a core candidate gene as its expression also increased in *N. brasiliensis* infection. Candidate genes were also identified that inhibit Th2 responses, T cell function and cytokine production, are associated with Treg suppression, and that play important roles in the development of immune tolerance (Fig 6C). ILT3 (*Lilrb4*) is expressed by Foxp3$^+$ Tregs during *L. sigmodontis* infection [37], and we confirmed that expression of ILT3 protein was significantly increased on the surface of d 60 CD4$^+$ Th2 cells (Fig 6D).

Granzyme B inhibits protective immunity to *L. sigmodontis* [38], and was upregulated by PleC Th2 cells between d 20 and 60 pi, both at the mRNA (13-fold increase, $p<9e^{-7}$), and protein levels (Fig 6E). However, Granzyme B was not represented in the core candidate gene list as it was also highly upregulated in *N. brasiliensis* Th2 cells compared to naive (37-fold, adj. $p < 1.6e^{-13}$). Furin is a pro-TGF-β1 converting enzyme [39], and TGF-β signalling genes (*Snx9* & *Skil*) were upregulated in the core candidate gene list. IPA upstream regulator analysis of the core PleC candidate genes, based on fold change between d 60 and d 20 pi, indicated that the TGF-β1 pathway was significantly enriched and predicted to be activated ($p = 1.58e^{-14}$, F = 2.0) (S6 Fig).

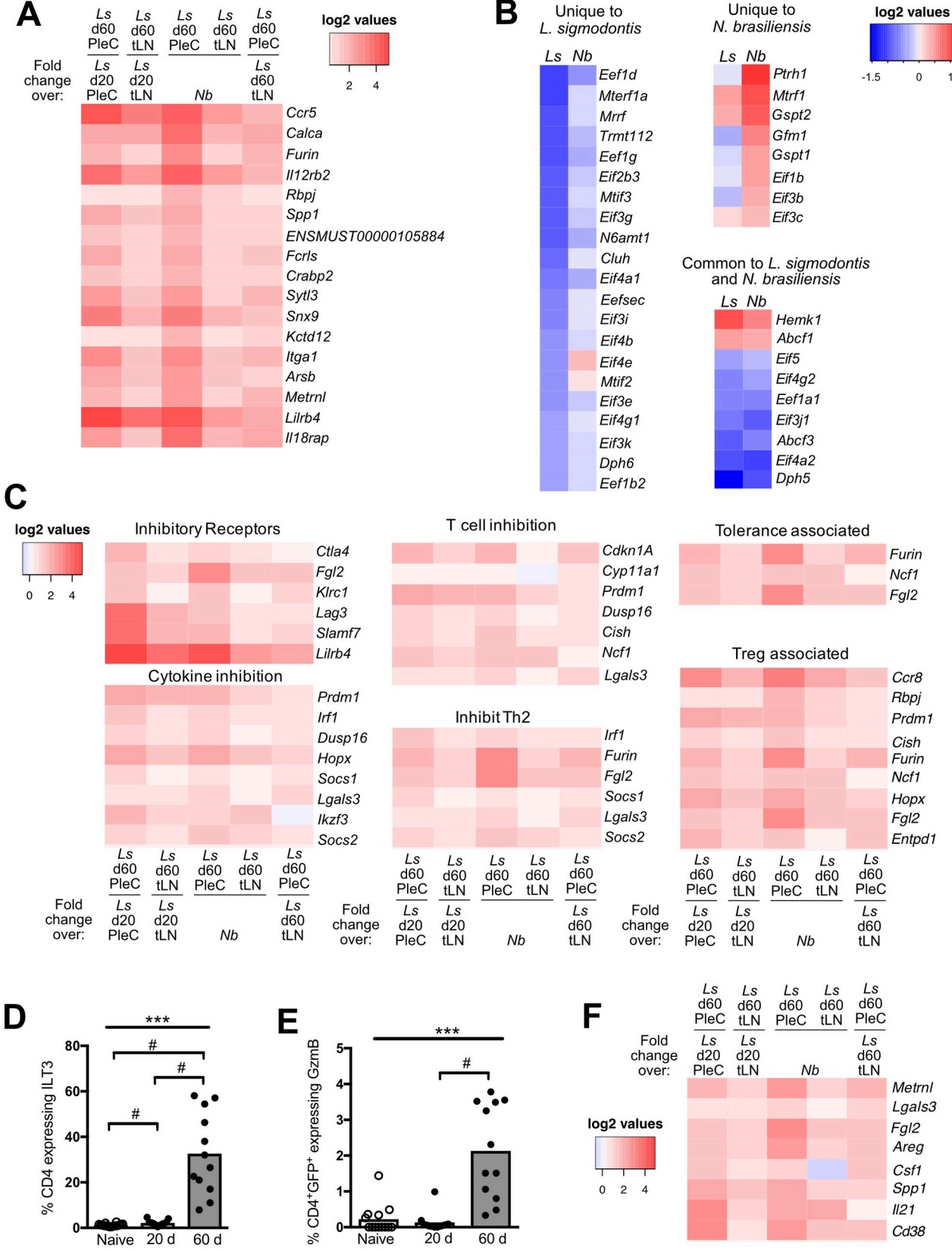

**Fig 6. Identification of genes associated with Th2 cell dysfunction.** (A) Heatmap showing fold change in expression of candidate genes that significantly differ between d 20 and d 60 of *L. sigmodontis* (*Ls*) infection in both the PleC and tLN, have significantly higher expression levels in the PleC versus tLN, and significantly differ from *N. brasiliensis (Nb)* Th2 cells. (B) Heatmap showing fold change of translation factors that uniquely or commonly significantly differ in *L. sigmodontis* PleC IL-4gfp⁺ Th2 cells and *N. brasiliensis* Th2 cells compared to naïve CD4⁺ T cells. (C) Heatmaps showing fold-change in expression of core candidate genes with known immune regulatory functions. Fold-change is calculated relative to Th2 cells from d 20 of *L. sigmodontis* infection, *N. brasiliensis* Th2 cells, and tLN Th2 cells from d 60 of *L. sigmodontis* infection. (D & E) Percentage of PleC CD4⁺ T cells expressing ILT3 (D) and CD4⁺IL-4gfp⁺ Th2 cells expressing Granzyme B (E) in naïve and *L. sigmodontis* infected mice at d 20 and d 60 pi. Graphs show combined data from two independent experiments. Symbols represent individual animals and bars show means. ***Significant effect between groups (p<0.001, ANOVA based on combined data from two independent experiments), # p<0.05 (Tukey HSD). (F) Heatmap showing fold-change in expression of core candidate genes with known secreted immune regulatory functions. Fold-change is calculated relative to Th2 cells from d 20 of *L. sigmodontis* infection, *N. brasiliensis* Th2 cells, and tLN Th2 cells from d 60 of *L. sigmodontis* infection.

The candidate gene list also contained a range of secreted factors with immune regulatory functions (Fig 6F). Of these, IPA identified CD38 (p = 3.4e⁻¹³), IL-21 (p = 4.93e-18), CSF1 (p = 2.25e⁻¹¹), Spp1 (p = 2.3e⁻⁴), and Lgals3 (p = 2.4e⁻⁶) as potential upstream regulators of the candidate genes, suggestive of autocrine activity. Furthermore, the CD38 (F = 3.3), CSF1 (F = 2.8) and IL-21 (F = 2.0) pathways were predicted to be activated. This suggests that intrinsically hypo-responsive Th2 cells are not functionally inert, and have developed alternate regulatory or effector functions.

## IL-21 inhibits protective and humoral immunity to *L. sigmodontis*

Our data suggests that, in contrast to their loss of ability to produce classical Th2 cytokines (IL-4, IL-5) [19], Th2 cells gain IL-21 production as they become hypo-responsive (Fig 6F). This switch from a Th2 response towards an IL-21 response indicated that IL-21 is playing a prominent role in regulating immunity to *L. sigmodontis* infection during chronicity. To confirm IL-21 expression at the protein level, we used flow cytometry to assess IL-21 production by CD4⁺ T cells at d 20 and 60 of *L. sigmodontis* infection. In agreement with the gene array data, PleC CD4⁺ T cells did not produce IL-21 during the active Th2 phase of infection (d 20 pi), but did produce IL-21 protein at d 60 pi, during the hypo-responsive phase of infection (Fig 7A & 7B).

To test the role of IL-21 during the hypo-responsive phase of *L. sigmodontis* infection, IL-21 activity was neutralised in WT BALB/c mice between days 45 and 60 of *L. sigmodontis* infection using an anti-IL-21R blocking antibody. IL-21R blockade increased resistance to *L. sigmodontis* infection to a similar extent as PD-1 blockade as it significantly reduced the levels of blood microfilaria without impacting the number of adult parasites recovered (Fig 7C) [19].

To determine whether IL-21 is involved in the maintenance of Th2 cell hypo-responsiveness, we assessed whether IL-21 blockade enhanced Th2 cell expansion or ability to produce cytokines. IL-21R blockade did not increase the number of CD4⁺GATA-3⁺ Th2 cells within the PleC (Fig 7D) or recover the ability of CD4⁺GATA-3⁺ Th2 cells to produce IL-4 or IL-5 protein (Fig 7E). This indicates that IL-21 does not maintain Th2 cell-intrinsic hypo-responsiveness, and that the increase in resistance was not due to restoration of Th2 cell function.

As IL-21 is typically associated with promoting humoral immunity [40], we assessed whether it regulated Tfh and B cell responses to *L. sigmodontis*. IL-21R blockade significantly increased the percentage of CD19⁺PNA⁺ germinal centre B cells (Fig 7F), as well as serum levels of *L. sigmodontis* specific IgG1 (Fig 7G). Although this did not translate to increased total numbers of CD19⁺PNA⁺ germinal centre B cells (Fig 7H), it suggests that IL-21 inhibits *L. sigmodontis* specific B cell responses. IL-21 blockade had no effect on the expansion of CD4⁺Bcl6⁺CXCR5ʰⁱ Tfh cells (Fig 7I), and did not change the numbers of CD19⁺IgM⁻IgD⁻ class-switched B cells (Fig 7J). Thus, IL-21 inhibits protective and humoral immunity to *L. sigmodontis*, but is not involved in the maintenance of Th2 cell-intrinsic hyporesponsiveness.

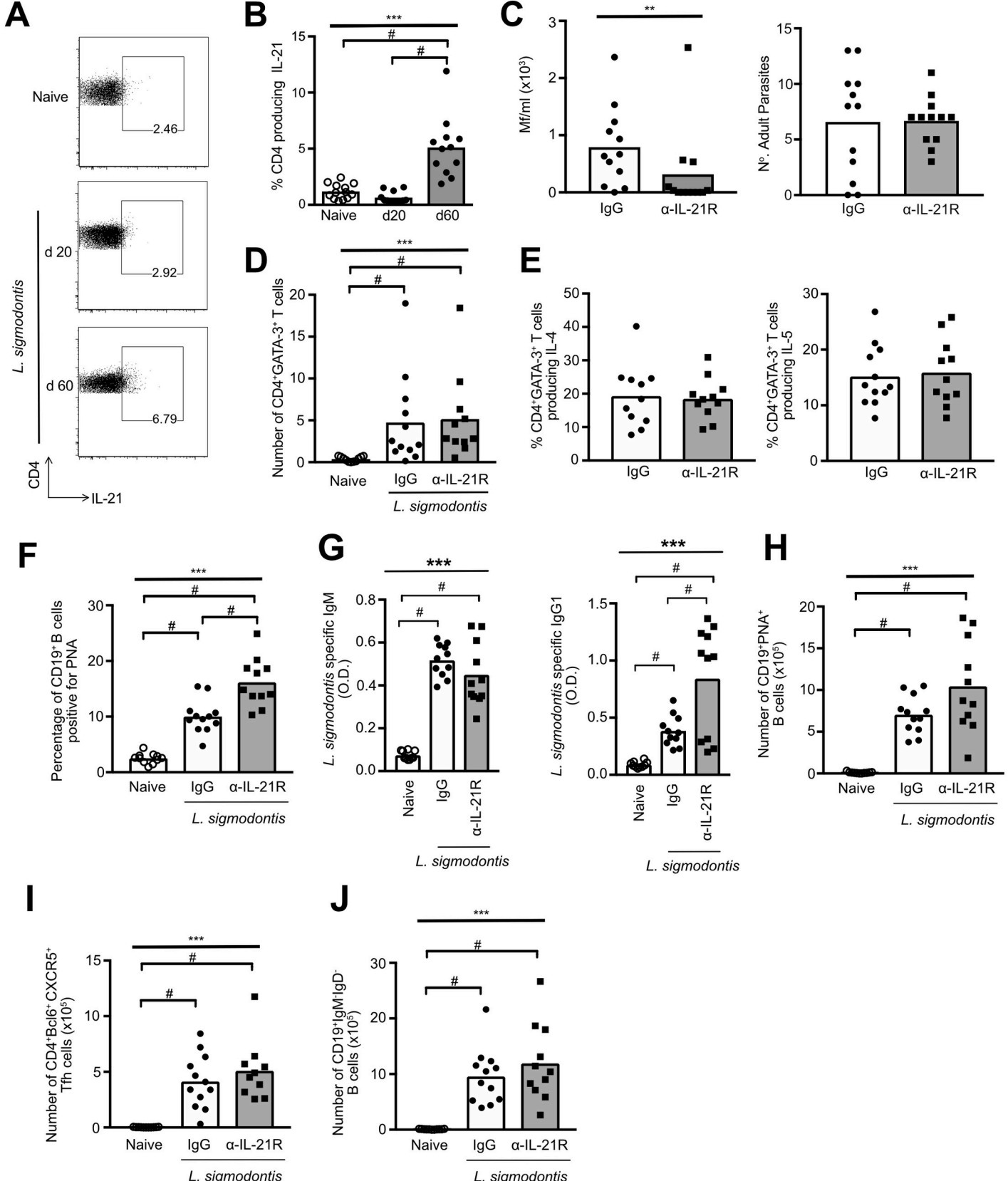

**Fig 7. IL-21R blockade increases resistance to *L. sigmodontis*.** (A) Representative flow staining showing IL-21 production by PleC CD4$^+$ T cells from naïve, and *L. sigmodontis* infected mice. (B) IL-21 protein production by PleC CD4$^+$ T cells from naïve and *L. sigmodontis* infected mice at d 20 and d 60 pi. Graph shows combined data from two independent experiments. Symbols represent individual animals and bars show means. $^{***}$Significant effect between groups (p<0.001, ANOVA using combined data from two independent experiments), # p<0.05 (Tukey HSD). (C–J) *L. sigmodontis* infected BALB/c mice were treated with a neutralising anti-IL-21R or rat IgG antibody between d 45 and 60 pi, and autopsied at d 60. Panels show combined data from two independent experiments with 5–6 mice per group. Symbols represent individual animals and bars show means. $^{**}$p < 0.01, $^{***}$ p < 0.001 (ANOVA using combined data from two independent experiments), # p < 0.05 (Tukeys HSD). (C) Number of Mf/ml of blood, and number of adult *L. sigmodontis* parasites recovered from the pleural cavity. (D) Number of PleC CD4$^+$GATA-3$^+$ T cells. (E) Percentage of PleC CD4$^+$GATA-3$^+$ T cells producing IL-4 or IL-5 protein. (F) Percentage of PNA$^+$CD19$^+$ B cells within the tLN. (G) Serum levels of *L. sigmodontis* specific IgM and IgG1. (H) Total number of PNA$^+$CD19$^+$ B cells within the tLN. (I) Total number of CD4$^+$Bcl6$^+$CXCR5$^+$ Tfh cells within the tLN. (J) Total number of CD19$^+$IgM$^-$IgD$^-$ class-switched B cells within the tLN.

## Discussion

T cell-intrinsic regulation, such as anergy, adaptive tolerance, exhaustion, and plasticity, is an important way in which immune responses are controlled. Whilst Th2 cell-intrinsic regulation is known to develop during helminth infections [11,16–19], and controls both resistance to infection and pathology [16,19], its phenotype is not well described. In this study we demonstrate that during *L. sigmodontis* infection the development of Th2 cell-intrinsic hypo-responsiveness associates with a switch from a classical IL-4$^+$IL-5$^+$IL-13$^+$ Th2 cell phenotype at d 20 pi, to a non-classical Egr2$^+$cMaf$^+$Blimp-1$^+$IL-4$^{lo}$IL-5$^{lo}$IL-13$^{lo}$IL-21$^+$T-bet$^+$IFN-γ$^+$ Th2 cell phenotype at d 60 pi. The d 60 hypo-responsive IL-4gfp$^+$ Th2 population had a divergent transcriptional profile to classical Th2 cells isolated at d 20 of *L. sigmodontis* infection or from mice acutely infected with *N. brasiliensis*. Transfer studies demonstrated that the hypo-responsive Th2 cells retained their dysfunctional phenotype in the absence of antigen and active infection. This indicates that Th2 cell-intrinsic hypo-responsiveness represents a distinct and stable state of Th2 cell differentiation.

Identifying shared traits of dysfunctional Th cells in diverse chronic settings is beneficial for the development of novel therapeutics. Our analysis is the first to compare Th2 cell-intrinsic dysfunction to known forms of T cell tolerance and unresponsiveness to find shared regulatory mechanisms. Hypo-responsive Th2 cells had dissimilar overall transcriptional profiles to exhausted, anergic, and tolerised CD4$^+$ T cells, indicating the development of a novel form of T cell dysfunction. In particular, showing an inverse gene expression pattern to exhausted CD4$^+$ T cells. As Th2 cell-intrinsic dysfunction is PD-1 dependent [19], this indicates that PD-1 co-inhibition leads to non-exhaustion forms of T cell dysfunction in Type 2 settings. Dysfunctional Th2 cells shared specific characteristics with these forms of T cell dysfunction, and compared to active d 20 Th2 cells had a more similar expression profile to anergic T cells. Hypo-responsive Th2 cells upregulated Blimp-1 and c-Maf transcription factors, which associate with inhibition of proliferation and increased IL-10 production respectively [41–43]. They also switched on the transcription factor Egr2, which plays a central role in the induction of T cell anergy [27,35], and upregulated expression of key genes and proteins associated with tolerance, including *Lag3*, *Tigit*, *Havcr2*, *Icos*, *Il10*, IL-21, c-Maf, *Nfil3*, *Ahr*, *Furin* and *Ncf1* [36,39,44]. Of these, *Furin* is involved in the activation of TGF-β from its latent form [39], and TGF-β was predicted to be an upstream regulator of the hypo-responsive Th2 cells suggesting a role in generating or maintaining the hypo-responsive state. Post-transcriptional silencing is a mechanism of shutting down cytokine production in anergic T cells [45], and *L. sigmodontis*-elicited hypo-responsive Th2 cells retained expression of mRNA for *Il4*, *Il5*, *Il13*, and *Il10* indicating that their downregulation is controlled at the translational level. Furthermore, Th2 cells from *L. sigmodontis* infected mice had reduced expression of translation factors, although their expression did not directly correlate with the onset of hypo-responsiveness.

A tolerised Th2 cell phenotype is consistent with published literature showing that human T cells display characteristics of anergy during helminth infections [13–15]. PBMC from

human filariasis patients upregulate anergy-associated ubiquitin ligases, including Nedd4 [15], while Th2 dysfunction in murine schistosomiasis is dependent upon GRAIL [16]. Both of these factors were increased in Th2 cells during *L. sigmodontis* infection, although their expression did not directly correlate with the development of hypo-responsiveness. Egr2 and GRAIL are linked with the development of an anergic-like T cell phenotype in Type 2 (*Fasciola hepatica*) and Type 1 (*Trypanosoma cruzi* and *Mycobacterium tuberculosis*) infection settings [17,46,47]. Thus, Th2 cell-intrinsic hypo-responsiveness shares characteristics with T cell dysfunction in humans and other infection settings. It also may share some common regulatory mechanisms with anergic and tolerised CD4$^+$ T cells, despite indications that they are discrete phenotypes.

Anergy and tolerance can stimulate tolerised-memory, and there is recent evidence that exhausted T cells develop memory [8–10,48]. *L. sigmodontis*-elicited hypo-responsive Th2 cells retained their dysfunctional phenotype for 2 weeks upon transfer to a naïve host, indicating that the phenotype is stable in the absence of antigen and extrinsic parasite and host immune regulatory factors. Whilst this is too short a time span to draw firm conclusions, it raises the potential for tolerised Th2 memory cells. The development of a tolerised Th2 memory response may be one element explaining why protective immunity to helminths takes so long to develop [49], and has implications for vaccination of previously infected individuals if vaccine-induced responses have to outcompete pre-established dysfunctional Th2 cells. It could also have beneficial applications, as the ability to induce long-term tolerance in committed allergen-specific Th2 cells would represent an ideal approach for the treatment of allergies.

Despite their dysfunctional Th2 phenotype, hypo-responsive Th2 cells were not functionally inert and upregulated expression of soluble regulatory factors including IL-21, IFN-γ, *Fgl2*, *Cd38*, *Spp1*, *Areg*, *Metrnl*, *Lgals3*, and *Csf1*. Fgl2 has immune suppressive functions, inhibiting immunity to *Echinococcus multilocularis* [50], and is associated with suppression by Treg and inhibitory Th2 cells [51,52]. The hypo-responsive Th2 cells upregulated *Lag3* mRNA, and Egr2 and Lag3 co-expression defines a population of IL-10$^+$Foxp3$^-$ Treg cells [53]. They also switched on IL-21 and c-Maf protein, which play central roles in the differentiation of Tr1 cells [42,43]. Thus, similar to *F. hepatica* anergised T cells that are inhibitory [18], hypo-responsive Th2 cells may have a regulatory phenotype.

IL-21 has previously been shown to promote Th2, Tfh, and B cell responses, counteract T cell exhaustion, and inhibit Foxp3$^+$ Treg function [40,43]. In contrast, blockade of the IL-21R during the hypo-responsive phase of *L. sigmodontis* infection increased resistance to infection and enhanced parasite-specific B cell and antibody responses. This indicates a novel role for IL-21 in suppressing Type 2 immune responses and humoral immunity during chronicity. The increased resistance was not associated with a reversal of Th2 cell dysfunction, indicating that IL-21 does not maintain Th2 cell intrinsic hypo-responsiveness, but rather has a downstream regulatory role. Whilst IL-21 can also be produced by Tfh cells, NK cells, and Foxp3$^+$ Tregs, this suggests that *L. sigmodontis*-elicited hypo-responsive Th2 cells may represent a type of Foxp3$^-$ Treg cell, potentially inhibiting protective Type 2 immunity via IL-21.

Mixed Th1/Th2 responses are seen in human filariasis patients [54], and IL-5 and IFN-γ synergise to kill *L. sigmodontis* parasites [55]. The development of T-bet$^+$IFN-γ$^+$ Th2 cells during *L. sigmodontis* infection indicates that the IFN-γ response may be partly due to the development of Th2/Th1 hybrid cells. Importantly, the dysfunctional Th2 phenotype was not due to a Th2 to Th1 switch as the loss of Th2 cytokine producing cells was greater than the gain in those producing IFN-γ. The hypo-responsive Th2 population also retains a dominant, albeit muted, Th2 cytokine signature upon *in vitro* antigenic stimulation [19,20]. IL-21 inhibits T-bet and Th1 differentiation [43], suggesting that the IL-21 secreting Th2 cells and Th2/Th1 hybrids are distinct. Thus, the d 60 hypo-responsive Th2 population may be comprised of

different subpopulations, including Th2/Th1 hybrid cells and dysfunctional Th2 cells, which have opposing protective and inhibitory roles respectively.

Th2 cells show heterogeneity between different Type 2 infectious and allergic settings [32], indicating that Th2 cell phenotype is context dependent. Our study shows that Th2 cell phenotype is also heterogeneous over time, and that Th2 cells can switch from a classical Th2 phenotype towards a transcriptionally distinct intrinsically hypo-responsive phenotype that is stably retained in the absence of antigen. We hypothesise that the hypo-responsive Th2 population elicited during *L. sigmodontis* infection comprises at least two different non-classical Th2 cell variants; a dysfunctional Egr2$^+$cMaf$^+$IL-21$^+$IL-4$^{lo}$IL-5$^{lo}$IL-10$^{lo}$IL-13$^{lo}$ population with similarities to tolerance and regulatory T cells, and an IFN-$\gamma^+$T-bet$^+$IL-4$^-$IL-5$^-$ Th2/Th1 hybrid population that retains mRNA for *Gata3*, *Il4* and *Il5*. Further studies are now needed to resolve these different subpopulations, and determine whether they represent forms of tolerance, alternative Th2 effector cells, or regulatory Th2 cells. This ability of Th2 cells to change phenotype over time has important implications for understanding the outcomes of helminth infections and allergies. It also raises the prospect of new approaches for therapeutically manipulating committed Th2 cells towards different fates, for example to tolerise allergen-specific Th2 cells.

## Supporting information

**S1 Fig. Representative flow plots showing sorting strategy and purities for the tLN and spleen.**
(PDF)

**S2 Fig. Transcription factors that are differentially expressed between d 20 and d 60 PleC IL-4gfp$^+$ Th2 cells during *L. sigmodontis* infection.**
(PDF)

**S3 Fig. Transcription factors that are expressed in common by d20 and d60 PleC IL-4gfp$^+$ Th2 cells during *L. sigmodontis* infection.**
(PDF)

**S4 Fig. IFN-$\gamma$ is an upstream regulator of PleC d 60 IL-4gfp$^+$ Th2 cells.**
(PDF)

**S5 Fig. Comparison of gene expression in peptide-induced tolerance and Th2 cell-intrinsic hypo-responsiveness.**
(PDF)

**S6 Fig. TGF-$\beta$ is an upstream regulator of core candidate genes.**
(PDF)

**S1 Table. Table showing KEGG pathways enriched in PleC and/or tLN IL-4gfp$^+$ Th2 cells at d 60 pi, but not at d 20 pi.**
(DOCX)

**S2 Table. Table showing fold-change in expression of genes significantly upregulated in anergic T cells that was used to perform the unbiased clustering represent by the heat map in Fig 5D.**
(XLSX)

**S3 Table. Table showing fold-change in expression of genes significantly upregulated in T cells following peptide-elicited tolerance that was used to perform the unbiased clustering represented in the heat map in S5 Fig.**
(XLSX)

**S4 Table. Table of loci that significantly differ between d 20 and 60 of *L. sigmodontis* infection in PleC IL-4gfp⁺ Th2 cells.**
(XLSX)

**S5 Table. Table of loci that significantly differ between d 20 and 60 of *L. sigmodontis* infection in tLN IL-4gfp⁺ Th2 cells.**
(XLSX)

**S6 Table. Table of core candidate loci associated with Th2 cell-intrinsic hypo-responsiveness.**
(XLSX)

**S7 Table. Table showing subset of the core candidate loci that have significantly different expression in tLN IL-4gfp⁺ Th2 cells between d20 and d60.**
(XLSX)

## Acknowledgments

We would like to thank Alison Fulton for experimental support and maintaining the *L. sigmodontis* lifecycle, Martin Waterfall and Central Bioresearch Services for excellent technical support, Rebecca Brownlie, Marta Trüb, Sharon Campbell and Carlos Minutti for their helping hands on big experimental days, Robert Salmond, Judith Allen, Dominik Rückerl, Rose Zamoyska and Rick Maizels for helpful discussions, and Margo Chase-Topping for advice on statistical analysis.

## Author Contributions

**Conceptualization:** Johanna A. Knipper, Matthew D. Taylor.

**Data curation:** Johanna A. Knipper, Alasdair Ivens.

**Formal analysis:** Johanna A. Knipper, Alasdair Ivens, Matthew D. Taylor.

**Funding acquisition:** Matthew D. Taylor.

**Investigation:** Johanna A. Knipper, Matthew D. Taylor.

**Methodology:** Johanna A. Knipper, Alasdair Ivens, Matthew D. Taylor.

**Supervision:** Matthew D. Taylor.

**Writing – original draft:** Johanna A. Knipper, Matthew D. Taylor.

**Writing – review & editing:** Johanna A. Knipper, Alasdair Ivens, Matthew D. Taylor.

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
