## [Decision Letter · Decision Letter 0]

16 Sep 2019

Dear Dr. Taylor:

Thank you very much for submitting your manuscript "Th2 cell-intrinsic dysfunction during chronic helminth infection is distinct from exhaustion and is maintained in the absence of antigen" (PNTD-D-19-01151) for review by PLOS Neglected Tropical Diseases. Your manuscript was fully evaluated at the editorial level and by independent peer reviewers. The reviewers appreciated the attention to an important topic but identified some aspects of the manuscript that should be improved.

We therefore ask you to modify the manuscript according to the review recommendations before we can consider your manuscript for acceptance. Your revisions should address the specific points made by each reviewer.

(1) A letter containing a detailed list of your responses to the review comments and a description of the changes you have made in the manuscript.

(2) Two versions of the manuscript: one with either highlights or tracked changes denoting where the text has been changed (uploaded as a "Revised Article with Changes Highlighted" file ); the other a clean version (uploaded as the article file).

(3) If available, a striking still image (a new image if one is available or an existing one from within your manuscript). If your manuscript is accepted for publication, this image may be featured on our website. Images should ideally be high resolution, eye-catching, single panel images; where one is available, please use 'add file' at the time of resubmission and select 'striking image' as the file type. 

Please provide a short caption, including credits, uploaded as a separate "Other" file. If your image is from someone other than yourself, please ensure that the artist has read and agreed to the terms and conditions of the Creative Commons Attribution License at http://journals.plos.org/plosntds/s/content-license (NOTE: we cannot publish copyrighted images). 

(4) Appropriate Figure Files 

Please remove all name and figure # text from your figure files upon submitting your revision. Please also take this time to check that your figures are of high resolution, which will improve both the editorial review process and help expedite your manuscript's publication should it be accepted. Please note that figures must have been originally created at 300dpi or higher. Do not manually increase the resolution of your files. For instructions on how to properly obtain high quality images, please review our Figure Guidelines, with examples at: http://journals.plos.org/plosntds/s/figures

While revising your submission, please upload your figure files to the Preflight Analysis and Conversion Engine (PACE) digital diagnostic tool, https://pacev2.apexcovantage.com/ PACE helps ensure that figures meet PLOS requirements. To use PACE, you must first register as a user. Then, login and navigate to the UPLOAD tab, where you will find detailed instructions on how to use the tool. If you encounter any issues or have any questions when using PACE, please email us at figures@plos.org.

We hope to receive your revised manuscript by Nov 15 2019 11:59PM. If you anticipate any delay in its return, we ask that you let us know the expected resubmission date by replying to this email.

To submit your revised files, please log in to https://www.editorialmanager.com/pntd/

Sincerely,

Benjamin L Makepeace

Associate Editor

Timothy Geary

Deputy Editor

Reviewer's Responses to Questions

**Key Review Criteria Required for Acceptance?**

**Methods**

-Are the objectives of the study clearly articulated with a clear testable hypothesis stated?

-Is the study design appropriate to address the stated objectives?

-Is the population clearly described and appropriate for the hypothesis being tested?

-Is the sample size sufficient to ensure adequate power to address the hypothesis being tested?

-Were correct statistical analysis used to support conclusions?

-Are there concerns about ethical or regulatory requirements being met?

Reviewer #1: This is a very well written manuscript that clearly states the hypothesis and the objectives of the study. The study design is well chosen and uses several state-of-the-art methods to address the objectives. The number of replicates used for the microarray analysis is sufficient and all immunological experiments were repeated with a sufficient number of animals. There are no concerns regarding the statisctial analysis used, ethics or regulatory requirements.

Reviewer #2: Objectives are clear and design is appropriate to address the stated objectives.

Statistical analyses were correct. Study was IACUC approved.

Reviewer #3: The objectives of the study are well articulated. There is no clear testable hypothesis stated but this does not seem appropriate for the type of work reported. The overall goal of understanding the characteristics of hypo-responsive T cells during chronic worm infection is clear. In terms of sample size, the numbers of animals used are appropriate and statistical analyses are correct to support the conclusion. There is no concern about ethical or regulatory requirements being met.

**Results**

-Does the analysis presented match the analysis plan?

-Are the results clearly and completely presented?

-Are the figures (Tables, Images) of sufficient quality for clarity?

Reviewer #1: Overall, the results are clearly and completely presented and the figures and tables are of good quality.

However, there are a few minor points that have to be addressed: 

- Line 557: Reference of Fig. 7A is misplaced here. 

- Legends for Fig. 6 and 7 have to be corrected. Descriptions of some of the data shown in those figures is missing/replaced in the legends. 

- Heatmaps shown in Fig. 5D and S6: Tables that summarize the gene expression changes indicated in the heatmaps of Fig. 5D and S6 should be provided as suppl. data. 

- Fig. 1D/Line 311-314: The bar indicating the statistically significant difference in Fig. 1D seems misplaced and not in accordance to the description given in the results section. 

- Line 322: This should state 2 weeks, as the infection took place 2 weeks after the cell transfer.

- Please expand the legend of Fig. S6 to explain the experimental design used for the peptide induced tolerance.

Reviewer #2: Analysis matches the analysis plan.

Results are clearly and completely presented (except for critiques provided below).

The figures are of sufficient quality.

Reviewer #3: The results are extremely well presented and the results section has a strong narrative that makes it easy to understand the logic to the experiments. All figures are of sufficient quality for clarity.

**Conclusions**

-Are the conclusions supported by the data presented?

-Are the limitations of analysis clearly described?

-Do the authors discuss how these data can be helpful to advance our understanding of the topic under study?

-Is public health relevance addressed?

Reviewer #1: With exception of the impact on the humoral immune responses the conclusions are supported by the presented data and the limitations of the study are presented in the discussion. 

In detail, I have following questions and concerns regarding the interpretation of tha data: 

- Fig. 7C: What is your average frequency of Mf positive animals during L.s. infection? Do you always observe a microfilaremia in almost all infected animals?

- Fig. 7D, E: Can the authors exclude that a lack of changes in the number of CD4+GATA3+ and IL-4 and IL-5 producing CD4+GATA3+ T cells by α-IL-21R blockage is not missed due to a belated blockade starting at 45dpi? Does 45dpi present a time point of acute or hyporesponsive infection & T cell function?

- Fig. 7G: Increase in L.s.-specific IgG1 in the α-IL-21R treated L.s. infected animals is very heterogeneous, which may be due to the pooling of 2 experiments. Was the increase in specific IgG1 in the depleted group consistent in both depletion experiments? Given the presented data, the statement that IL-21 inhibits L.s.-specific B cell responses/humoral responses is not well supported and should be confirmed or restated throughout the manuscript (e.g. l. 594, l. 686-688, abstract).

- Are there any indications that epigenetic changes may occur? The possible contribution of epigenetic changes should be discussed. 

- Other potential sources of IL-21 and their possible contribution should be mentioned in the discussion

Reviewer #2: Yes, the conclusions are supported by the data. The authors do not overstate their findings. Limitations are described. Potential relevance is discussed.

Reviewer #3: The conclusions are supported by the data presented. There are definite limitations to the data presented, in particular the inability of the group to co-stain IL-4gfp with transcription factors that make it difficult to definitively define the phenotype of the hyporesponsive cells. However, these limitations are well discussed. The authors clearly discuss how the data may advance our understanding of T cell hypo-responsiveness and in particular discuss how this may be relevant to the publich health concerns of filarial nematode infection.

**Editorial and Data Presentation Modifications?**

Reviewer #1: (No Response)

Reviewer #2: Minor revision. No additional experiments needed. See notes below.

Reviewer #3: (No Response)

**Summary and General Comments**

Reviewer #1: This is a very well presented study that investigated the fate of Th2 cells during acute and chronic helminth infection. Interestingly, Th2 cells developed during chronic filarial infection a transcriptional profile that partially overlaps with anergic and tolerized T cells and differs from Th2 cells isolated during acute filarial or intestinal helminth infection. Furthermore, the hypo-responsive T cell phenotype was stable after transfer into naïve mice. While losing their ability to produce IL-4 and IL-5 proteins, the hypo-responsive T cells produced IL-21, which is suggested to support the development of microfilaremia.

The authors used to different models of helminth infections, microarray analysis and a thorough bioinformatic analysis of their gene expression data, which was in part confirmed by flow cytometry. Thus, the study represents a very robust work that further enhances our understanding of the immunomodulation of helminths and especially the development of hypo-responsive T cells.

Reviewer #2: SUMMARY/OVERVIEW

In this paper the investigators study changes in the transcriptome of dysfunctional Th2 cells that arise during chronic mouse infection with the filarial nematode Litomosoides sigmodontis.

Key findings include: showing that dysfunctional Th2 cells:

1) remain dysfunctional stably when transferred into a non-infected mouse

2) have a post-transciptional block for production of Th2 cytokines

3) have a divergent transcriptional profile when compared to classical Th2 cells that are isolated before hyporesponsiveness occurs in Litomosoides sigmodontis or after acute Nippostrongylus infection

4) upregulate Blimp-1 expression but have a transcriptional profile that is dissimilar from exhausted CD4 T cells

5) have a transcriptional profile that has some overlap with anergic/tolerized CD4 T cells, including upregulation of Egr2 and c-Maf

6) gain the ability to produce IL-21, which seems to be involved in a novel regulatory role of B cell antibody production during filarial infection.

The authors also found that there are many parasite-specfic differences in CD4 T cell transcriptional changes.

The results provide insights into how Th2 cells become dysfunctional in the context of chronic filarial infection. A strength is the careful comparison to transcriptional profiles associated with T cell exhaustion and T cell anergy, which clearly shows that what is occurring in Th2 cells during filarial infection is somewhat novel from either of these mechanisms of T cell dysfunction. Overall the methods and statistical analyses appear robust, and I suspect the article would be of interest to many researchers that study immune regulation during chronic helminth infection.

MAJOR CRITIQUES

1. Figure 7h I is supposed to be the total # of CD19+PNA+ germinal center B cells, but the figure is an exact copy of 7J, which is the # of CD19 Igm-IgD- class-switched B cells. Please correct.

MINOR CRITIQUES

1. In the results section, the authors state that d 60 CD4+IL-4gfp+ Th2 cells had “significantly impaired ability to produce IL-5 protein compared to d 20 CD4+IL-4gfp+ Th2 cells” in unchallenged mice. However, when looking at the datapoints in the figure, it doesn’t really appear as though there would be a statistically significant difference between those two groups. Additionally, the authors put a large bar with two asterisks above it across the entire figure. The authors in the results, however, appear to be comparing d20 to d60 in unchallenged and then in challenged mice….suggesting that perhaps they intended to include two bars in the figure (one over the first two groups, one over the final two groups).

2. In discussions of figure 3 in the text the authors switch from Tbx21 (which is in the figures) to using Tbet in the written results. This is ok, but a bit confusing to the reader…perhaps clarify with a parenthetical that Tbet is the same as Tbx21.

3. At times it is difficult to follow the logic for which genes are being assessed as a group. This was especially true for the results section in Figure 6. The authors may want to break up the results section covering Figure 6 into more paragraphs and provide a sentence or two explaining the groupings prior to discussing each individual figure in Figure 6.

4. The results section “Th2 cell-intrinsic hypo-responsiveness has similarities with T cell anergy and tolerance” has a concluding sentence “However, the overall transcriptional profile Th2 cell-intrinsic hyporesponsiveness was dissimilar to T cell anergy and tolerance” that contradicts the title. Would suggest simply changing the concluding sentence to something along the lines of “there is some overlap.”

5. Figure 7F y axis delete (x10exp5) from the axis title as it is percentage of B cells, not total numbers, being expressed.

Reviewer #3: In this manuscript by Knipper et al., the authors use an experimental filarial nematode infection (Litomosoides sigmodontis) to investigate the characteristics of hyporesponsive Th2 cells in this setting. Although, the authors do not directly study human filarial infection these findings may have relevance to human filarial disease as a similar hypo-responsive phenotype of T cells has also been described. Importantly, the data suggests that in chronic filarial infection Th2 cells switch from an active IL-4+IL-5+ Th2 phenotype to a modified and dysfunctional phenotype that is distinct from that described in other settings of T cell hyporesponsiveness (e.g. T cell exhaustion). These modified T cells are identified by their IL-21+Egr2+c-Maf+Blimp-1+IL-4loIL-5loTbet+IFN-gamma+ state. These may provide new therapeutic targets in filarial infection.

Overall, this is a novel and interesting study that provides new insight into T cell hyporesponsiveness during filarial nematode infection. A particular strength of the study is that there is excellent integration of microarray data with ex vivo flow cytometry data to define the global characteristics of hypo-responsive Th2 cells. Moreover, based on these findings the authors identify IL-21 as a cytokine involved in augmenting blood microfilarial levels at the chronic stage of L. sigmodontis infection.

One limitation to the study is the incapacity of the group to co-stain IL-4-gfp with transcription factors making it difficult to definitively demonstrate that the IL-4-gfp +ve cells (i.e. Th2 cells) are the ones that also co-express T-bet. However, this limitation is well discussed in the manuscript.

PLOS authors have the option to publish the peer review history of their article (what does this mean?). If published, this will include your full peer review and any attached files.

Reviewer #1: No

Reviewer #2: Yes: Edward Mitre, M.D. and Emilie Goguet, Ph.D.

Reviewer #3: Yes: Dr. John R. Grainger

---

## [Editor Report · Decision Letter 1]

6 Nov 2019

Dear Dr. Taylor,

We are pleased to inform you that your manuscript, "Th2 cell-intrinsic dysfunction during chronic helminth infection is distinct from exhaustion and is maintained in the absence of antigen", has been editorially accepted for publication at PLOS Neglected Tropical Diseases.

Before your manuscript can be formally accepted and sent to production you will need to complete our formatting changes, which you will receive in a follow up email. Please note: your manuscript will not be scheduled for publication until you have made the required changes.

IMPORTANT NOTES

* Copyediting and Author Proofs: To ensure prompt publication, your manuscript will NOT be subject to detailed copyediting and you will NOT receive a typeset proof for review. The corresponding author will have one final opportunity to correct any errors when sent the requests mentioned above. Please review this version of your manuscript for any errors.

* If you or your institution will be preparing press materials for this manuscript, please inform our press team in advance at plosntds@plos.org. If you need to know your paper's publication date for media purposes, you must coordinate with our press team, and your manuscript will remain under a strict press embargo until the publication date and time. PLOS NTDs may choose to issue a press release for your article. If there is anything that the journal should know, please get in touch.

*Now that your manuscript has been provisionally accepted, please log into EM and update your profile. Go to http://www.editorialmanager.com/pntd, log in, and click on the "Update My Information" link at the top of the page. Please update your user information to ensure an efficient production and billing process.

*Note to LaTeX users only - Our staff will ask you to upload a TEX file in addition to the PDF before the paper can be sent to typesetting, so please carefully review our Latex Guidelines [http://www.plosntds.org/static/latexGuidelines.action] in the meantime.

Best regards,

Benjamin L Makepeace

Associate Editor

Timothy Geary

Deputy Editor

---

## [Editor Report · Acceptance letter]

4 Dec 2019

Dear Dr. Taylor,

We are delighted to inform you that your manuscript, "Helminth-induced Th2 cell dysfunction is distinct from exhaustion and is maintained in the absence of antigen," has been formally accepted for publication in PLOS Neglected Tropical Diseases.

Best regards,

Serap Aksoy

Editor-in-Chief

Shaden Kamhawi

Editor-in-Chief
